# Exploring Nonlinear Pathway in Parameter Space for Machine Unlearning

**Yingdan Shi** [1]   **Ren Wang** [1]

## Abstract

Machine Unlearning (MU) aims to remove the information of specific training data from a trained model, ensuring compliance with privacy regulations and user requests. While one line of existing MU methods relies on linear parameter updates via task arithmetic, they suffer from weight entanglement. In this work, we propose a novel MU framework called **M**ode **C**onnectivity **U**nlearning (MCU) that leverages mode connectivity to find an unlearning pathway in a nonlinear manner. To further enhance performance and efficiency, we introduce a parameter mask strategy that not only improves unlearning effectiveness but also reduces computational overhead. Moreover, we propose an adaptive adjustment strategy for our unlearning penalty coefficient to adaptively balance forgetting quality and predictive performance during training, eliminating the need for empirical hyperparameter tuning. Unlike traditional MU methods that identify only a single unlearning model, MCU uncovers a spectrum of unlearning models along the pathway. Overall, MCU serves as a plug-and-play framework that seamlessly integrates with any existing MU methods, consistently improving unlearning efficacy. Extensive experiments on the image classification task demonstrate that MCU achieves superior performance. The code is available at https://github.com/TIML-Group/Mode-Connectivity-Unlearning.

## 1. Introduction

Machine Unlearning (MU) has emerged as a critical capability to comply with privacy regulations and user-initiated data removal requests. The most straightforward way is to remove the forgetting data and then train the model from scratch. However, this retraining method demands substantial computational overhead. To address this issue, various approximate MU methods (Fan et al., 2023; Graves et al., 2021; Ilharco et al., 2022; Kurmanji et al., 2024a; Thudi et al., 2022; Warnecke et al., 2021; Sun et al., 2024) have emerged to provide a more efficient alternative through diverse techniques.

A mainstream MU approach uses *linear* negation task arithmetic (Ilharco et al., 2022; Ortiz-Jimenez et al., 2024), where the unlearning model is obtained by linearly subtracting the forgetting task vector from the original model. However, modern classifiers exhibit a high complexity of high-dimensional representation and nonlinear characteristics, where simple linear updates may fail to remove forgetting information exclusively without introducing side effects. Specifically, **linear task arithmetic suffers from weight entanglement**, as the task vectors fail to localize their influence solely to the forgetting data without interfering with others, which is a violation of the necessary condition for successful linear editing (Ortiz-Jimenez et al., 2024). The detailed theoretical proof is provided in the Appendix A. Thus, this raises an important question as follows:

> **(Q1)** *Can we break free from the constraints of linear updates and instead explore MU in a nonlinear manner?*

If an alternative nonlinear pathway is uncovered, it can offer a more effective unlearning without side effects from weight entanglement. Another limitation of existing MU methods is that they typically yield a *single* unlearning model. Existing work (Georgiev et al., 2024) shows that **the optimal stopping point varies across different forgetting data, and therefore a single unlearning model is inherently incapable of simultaneously achieving effective unlearning for all forgetting points**. In contrast, exploring an unlearning pathway provides a promising solution to the limitations inherent in a single model (see Appendix B for proof). Thus, the other question arises:

> **(Q2)** *Can we identify a spectrum of effective MU models rather than just one?*

**A spectrum of effective unlearning models would also enable us to select the solution that best aligns with specific priority, such as prioritizing model utility preservation**

[1]Department of Electrical and Computer Engineering, Illinois Institute of Technology, Chicago, IL, USA. Correspondence to: Ren Wang <rwang74@iit.edu>.

*Proceedings of the 43rd International Conference on Machine Learning*, Seoul, South Korea. PMLR 306, 2026. Copyright 2026 by the author(s).

**or forgetting quality without repeated training.** For instance, in harmful information removal, the perceived risk of a data sample may evolve over time. A sample once considered low-risk may later be deemed highly risky, requiring stronger forgetting. Thus, exploring unlearning pathway provides greater flexibility in practical applications without requiring costly recomputations of different solutions.

To address these questions, we propose to explore unlearning pathway in the parameter space in a *nonlinear* manner, inspired by mode connectivity (Garipov et al., 2018). Our main contributions are summarized as follows:

- We identify the weight entanglement issue in existing linear unlearning methods and, for the first time, investigate unlearning from a nonlinear perspective.

- We introduce the novel concept of exploring unlearning pathways, opening a new direction for unlearning.

- As a plug-and-play framework, our approach can be seamlessly integrated with existing unlearning methods to effectively enhance their performance, mitigating both over-forgetting and under-forgetting issues.

- We show that masking entire parameters can achieve comparable effectiveness in unlearning the pathway while significantly reducing training time compared to existing masking approaches.

## 2. Related Work

### 2.1. Machine Unlearning

Retraining for MU involves retraining from scratch after removing forgetting data, but its high cost has led to the development of efficient approximate unlearning techniques. Some works (Fan et al., 2023; Graves et al., 2021; Kurmanji et al., 2024a; Shi & Wang, 2025; Tarun et al., 2023; Thudi et al., 2022) focus on designing loss functions to achieve forgetting. Knowledge distillation-based methods (Chundawat et al., 2023a;b; Goel et al., 2022; Kurmanji et al., 2024b; Micaelli & Storkey, 2019) train a student model to mimic the behavior of the original model on the retaining dataset while excluding the forgetting data. Several works (Foster et al., 2024; Golatkar et al., 2020; Liu et al., 2023) leverage the Fisher Information Matrix to identify and modify the most influential parameters associated with the forgetting data, enabling more targeted and efficient unlearning. Additionally, adversarial attacks (Cha et al., 2024; Chen et al., 2021; Wei et al., 2023) and differential privacy (Guo et al., 2019; Huang & Canonne, 2023) have also been explored as promising techniques for MU.

One pivotal advance came from task arithmetic (Ilharco et al., 2022), which enabled efficient data removal via negation operations. Building on this, a neural tangent kernel-based linear negation method was introduced to improve task arithmetic by constraining model updates to the tangent space (Ortiz-Jimenez et al., 2024). However, the entanglement issue still exists as they cannot guarantee that the task vector's influence localizes solely on forgetting data (see Appendix A for details). Overall, the oversimplified assumption of linear parameter updating fails to account for the nonlinear characteristics of loss landscapes and suffers from a weight entanglement issue.

A concurrent work (Cheng & Amiri, 2025) proposes a mode connectivity framework for assessing whether different unlearning strategies converge to mechanistically similar solutions. Their approach primarily relies on mode connectivity as an unlearning evaluation tool. In contrast, our work leverages mode connectivity as a core component of the unlearning procedure itself and further addresses the weight entanglement issue inherent in linear unlearning methods.

### 2.2. Mode Connectivity

Mode connectivity denotes low-loss pathways between different local minima in the loss landscape. It has been observed that neural networks with different initializations can be connected by a smooth, low-loss curve in parameter space (Garipov et al., 2018). This phenomenon has been further explored, demonstrating that such connectivity generalizes across architectures and datasets, forming high-dimensional manifolds of functionally equivalent models (Draxler et al., 2018). Recent work (Ren et al.) extends the mode connectivity from the Bézier curve to surface, and some other works utilize the property for evaluating or enhancing model robustness (Wang et al., 2023; 2024; Kim et al., 2026b; Zhang et al., 2026; Kim et al., 2026a). Given its ability to identify meaningful pathways in parameter space, mode connectivity provides an efficient and effective approach for unlearning.

## 3. Mode Connectivity Unlearning

### 3.1. Preliminaries and Notations

In the context of MU for image classification, we consider two scenarios: **random data forgetting** and **class-wise forgetting**. In random data forgetting, a subset of training data is randomly selected to form the forgetting data. In class-wise forgetting, the training data belonging to a specific class is designated as forgetting data. Let $\mathcal{D}_{train}$ be the full training dataset and $\mathcal{D}_v$ be the validation dataset. We use $\mathcal{D}_f \in \mathcal{D}_{train}$ to denote the forgetting data and $\mathcal{D}_r = \mathcal{D}_{train} \setminus \mathcal{D}_f$ to denote the retaining data. The test data is denoted as $\mathcal{D}_t$. In the class-wise scenario, $\mathcal{D}_t = \mathcal{D}_{tr} \cup \mathcal{D}_{tf}$ where $\mathcal{D}_{tr}$ and $\mathcal{D}_{tf}$ are test-retaining data and test-forgetting data respectively. The objective of our work is to identify a pathway, where each point along the pathway corresponds to an unlearning model, denoted by $\boldsymbol{\theta}_u$.

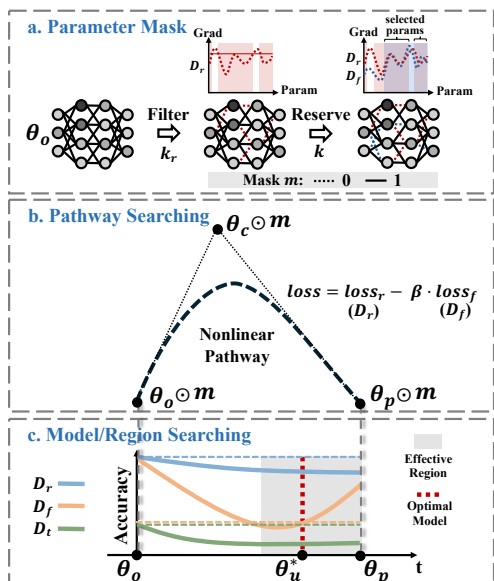

*Figure 1.* Overview of MCU framework. **a.** Identify a parameter mask by excluding the top-$k_r$ parameters important for retaining data, then preserving the top-$k$ parameters critical for forgetting data. **b.** Explore nonlinear pathways in the parameter space, where $\theta_c$ serves as the control point. **c.** Locate the optimal unlearning model and effective unlearning region along the pathway.

### 3.2. Unlearning Pathway Searching

Figure 1 shows the overview of our Mode Connectivity Unlearning framework **MCU**. As shown in Figure 1b, one crucial decision is the selection of two end models on the pathway. Ideally, these two end models should satisfy the following properties for unlearning: ① One end model should fully preserve model utility; ② The other end model can provide essential unlearning information and trend. Then we can find an optimal pathway between two end models, ensuring a balance between model utility and unlearning effectiveness. Guided by these insights, we define two specific models as two end models in our nonlinear pathway:

① **Original model** $\theta_o$ is trained on the training data $\mathcal{D}_{train}$ before unlearning.

② **Pre-unlearning model** $\theta_p$ is obtained by applying any existing MU method to remove the influence of forgetting data $\mathcal{D}_f$. $\theta_p$ already encodes a directional shift in parameter space toward forgetting relative to $\theta_o$. It captures a coarse but informative trajectory of how the model parameters should change to reduce the influence of $\mathcal{D}_f$.

The goal of MCU is to construct a smooth pathway from $\theta_o$ to $\theta_p$, ensuring an unlearning model $\theta_u$ on the pathway can better forgets $\mathcal{D}_f$ while preserving performance on $\mathcal{D}_r$. **As a plug-and-play framework that builds upon existing unlearning methods, the performance of MCU may exhibit some fluctuations due to different $\theta_p$. However, MCU can consistently identify improved unlearning models along the pathway regardless of $\theta_p$ quality.** Inspired

by (Garipov et al., 2018), we leverage a quadratic Bézier curve as our default setting to explore a nonlinear unlearning pathway in the parameter space. For comparison, we also present the results on Polychain in Table 4 and Figure 8, and the results on linear interpolation in Table 9. In our MU scenario, the quadratic Bézier curve $\phi_{\theta}(t)$ between models $\theta_o$ and $\theta_p$ in parameter space is defined as follows,

$$\phi_{\theta_c}(t) = (1-t)^2 \theta_o + 2(1-t)t\theta_c + t^2 \theta_p, \quad t \in [0, 1], \quad (1)$$

where $\theta_c$ is the control model, and $t$ represents a scalar interpolation coefficient that controls the position along the pathway connecting two end models in the high-dimensional parameter space. $\phi_{\theta_c}(t)$ parameterized by coefficient $t$ represents a continuous Bézier curve that smoothly transitions between models $\theta_o$ and $\theta_p$. As $t$ varies within the range $[0, 1]$, $\phi_{\theta_c}(0) = \theta_o$ corresponding to the original model and $\phi_{\theta_c}(1) = \theta_p$ corresponding to the pre-unlearning model. For values of $t$ between 0 and 1, it represents a spectrum of potential unlearning models $\theta_u$ along the pathway.

The control model $\theta_c$ in Eq. 1 serves to shape the Bézier curve. By optimizing this control model, we can influence the trajectory between $\theta_o$ and $\theta_p$. However, simply constructing a smooth path is insufficient for effective unlearning. It is therefore crucial to design an appropriate loss function that guides the optimization of the control model. This loss must strike a balance between two goals, ensuring effective forgetting and preserving model utility. This leads to our loss design,

$$\mathcal{L}_{mcu} = \mathbb{E}_{t \sim U(0,1)}[\mathcal{L}(\mathcal{D}_r; \phi_{\theta_c}(t)) - \beta \cdot \mathcal{L}(\mathcal{D}_f; \phi_{\theta_c}(t))], \quad (2)$$

where $\mathcal{L}(\mathcal{D}_r; \phi_{\theta_c}(t))$ is the cross-entropy loss on retaining data $\mathcal{D}_r$, and $\beta$ is an unlearning penalty coefficient controlling the trade-off between retaining predictive performance and forgetting quality. $U(0, 1)$ is the uniform distribution on $[0, 1]$, from which we sample a value $t$ for each training batch following the work (Garipov et al., 2018). In each batch, the loss is computed at the specific point $\phi_{\theta_c}(t)$ along the pathway, derive gradients with respect to $\theta_c$, and update only $\theta_c$ accordingly. Note that the pathway searching process only requires optimizing $\theta_c$, while the entire pathway is a simple combination of $\theta_o$, $\theta_c$ and $\theta_p$ as defined in Eq. 1.

Based on the issue proposed by (Georgiev et al., 2024), we establish that relying on a single model is fundamentally insufficient due to the misalignment of optimal stopping epochs across different forgetting data:

**Theorem 3.1.** *(Informal) Assume the unlearning procedure consists of $T$ training epochs, and let $|\mathcal{D}_f|$ denote the number of forgetting data points. For any confidence level $1 - \delta \in (0, 1]$, achieving probability at least $1 - \delta$ that each forgetting data point is optimally unlearned requires at least $k(\delta) = \lceil \frac{\ln \delta}{\ln(1 - T^{-|\mathcal{D}_f|+1})} \rceil$ distinct models. See Appendix B for the formal version and proof.*

For the case of $T = 10$, $|\mathcal{D}_f| = 100$, and $\delta = 0.05$, we obtain $k(0.05) \approx 2.996 \times 10^{99}$, which is an astronomically large number. This implies that achieving a high-probability guarantee requires training at least $k(\delta)$ distinct models, which is computationally prohibitive and practically infeasible. In contrast, our proposed unlearning pathway generates a continuous spectrum of unlearning models along a parameterized trajectory, providing an efficient and elegant resolution to this challenge.

### 3.3. Parameter Mask

While the above pathway search is efficient, we further improve efficiency by selectively updating parameters. Existing element-level parameter mask approaches (Fan et al., 2023; Huang et al., 2025) already show the effectiveness for preserving retaining performance and enhancing forgetting quality. However, in the element-level parameter mask, gradient computations are still required for all parameters, which limits practical efficiency gains. Here, **we show that masking an entire parameter achieves comparable effectiveness**, enabling computational speedup by completely bypassing gradient computations for the masked parameters.

As illustrated in Figure 1a, our parameter mask strategy consists of two key components: filtering based on $\mathcal{D}_r$ and reserving based on $\mathcal{D}_f$. The strategy effectively identifies parameters that are highly influential for $\mathcal{D}_f$ while being less critical for $\mathcal{D}_r$, ensuring a more targeted update process.

**Filtering based on $\mathcal{D}_r$.** We first utilize the gradient of the retaining loss with respect to the original model $\boldsymbol{\theta}_o$ on the retaining dataset $\mathcal{D}_r$. A fraction $k_r$ of the parameters is selected for exclusion, where these parameters exhibit an importance above a quantile-based threshold $\gamma_{k_r}$. The formulated equation is as follows,

$$\boldsymbol{m}_r^i = \mathbb{0}\left\{\frac{\|\nabla_{\boldsymbol{\theta}_o^i}\mathcal{L}(\mathcal{D}_r;\boldsymbol{\theta}_o)\|_2}{|\boldsymbol{\theta}_o^i|} > \gamma_{k_r}\right\}, \qquad (3)$$

where $\boldsymbol{m}_r^i$ is the binary mask for the $i$-th parameter in whole mask $\boldsymbol{m}$, and $\|\cdot\|_2$ denotes the $L_2$-norm over each parameter. $L_2$-norm reflects the Euclidean length of gradient vectors, making it more sensitive to parameters with larger impacts. The denominator $|\boldsymbol{\theta}_o^i|$ represents the element number in the entire $i$-th parameter of $\boldsymbol{\theta}_o$, i.e., $\boldsymbol{\theta}_o^i$, ensuring fair importance calculation across parameters with different sizes. The indicator function $\mathbb{0}(\cdot > \gamma_{k_r})$ assigns a zero vector to $\boldsymbol{m}_r^i$ if the average importance of this parameter exceeds the threshold $\gamma_{k_r}$, and otherwise an all-ones vector.

**Reserving based on $\mathcal{D}_f$.** After filtering, which removes parameters that are highly influential for $\mathcal{D}_r$, we further refine the mask by selecting parameters based on the gradient of the forgetting loss,

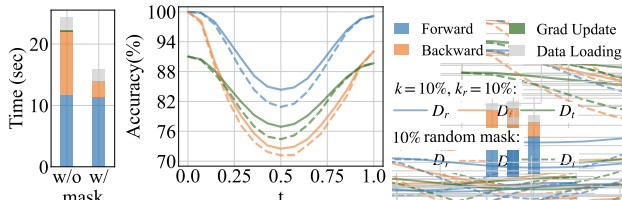

*Figure 2.* The efficiency and effectiveness of our parameter mask. 'w/o' and 'w/' in the left panel represent the results without 10% mask and with 10% mask. The x-axis of the right panel represents the parameter $t$ along the Bézier curve, while the y-axis corresponds to accuracy.

$$\boldsymbol{m}_f^i = \mathbb{1}\left\{\frac{\|\nabla_{\boldsymbol{\theta}_o^i}\mathcal{L}(\mathcal{D}_f;\boldsymbol{\theta}_o)\|_2}{|\boldsymbol{\theta}_o^i|} > \gamma_k\right\}. \qquad (4)$$

Similarly, the threshold $\gamma_k$ is determined by selecting the top-$k$ percentile of normalized gradient $L_2$ norms across parameters. The indicator function $\mathbb{1}(\cdot > \gamma_k)$ assigns an all-one vector to the entire $i$-th parameter $\boldsymbol{\theta}_o^i$ if its importance exceeds the threshold $\gamma_k$. Overall, our parameter mask is applied at the level of parameter tensors, each associated with a named parameter in the model. These typically correspond to components within submodules, such as the weights of a convolutional kernel or an attention projection matrix. A detailed experimental comparison between our masking strategy and existing element-wise masking strategies is provided in Figure 11 in Appendix D.

The final mask $\boldsymbol{m}$ is represented as,

$$\boldsymbol{m} = \boldsymbol{m}_r \,\&\, \boldsymbol{m}_f, \qquad (5)$$

where the operator $\&$ represents the logical operation AND. The parameter mask ensures that updates are restricted to the selected parameters, preventing unnecessary modifications to the model. Thus, the optimization in the MCU can be formulated as follows,

$$\min_{\boldsymbol{\theta}_c \odot \boldsymbol{m}} \mathcal{L}_{mcu}, \qquad (6)$$

where training efficiency is improved by reducing unnecessary gradient updates with the mask $\boldsymbol{m}$.

We preliminarily explore the efficiency and effectiveness of our parameter mask on CIFAR-10 with PreResNet-110 in the 10% random data forgetting. We set $k = 10\%$, $k_r = 10\%$ to generate our parameter mask with Neg-Grad+ (Kurmanji et al., 2024a) as the pre-unlearning model. In the left panel of Figure 2, we compare the average epoch runtime with and without our parameter mask. Apparently, the parameter mask significantly improves efficiency during the backward propagation process, achieving a notable 75.23% speedup. In the right panel of Figure 2, we compare our parameter mask (solid line) with the 10% random mask (dashed line). Random mask has a significant negative impact on the accuracy of $\mathcal{D}_r$ and $\mathcal{D}_t$, with a 3.44% and a 2.48% drop at $t = 0.5$ respectively. By comparison, the forgetting accuracy gap is only 1.32% at $t = 0.5$. This

confirms that our parameter mask both improves training efficiency and effectively preserves the model utility.

## 3.4. Adaptive Unlearning Penalty Coefficient

Through numerous experiments, our MCU with a fixed coefficient $\beta$ is good enough and can be empirically selected with ease. However, non-expert users may find it challenging to adjust this hyperparameter appropriately. Therefore, in the absence of prior experience, implementing an adaptive strategy for $\beta$ can avoid trial-and-error cost of hyperparameter selection and potentially improve our performance. As shown in the optimization objective (Eq. 2), balancing retaining ($\mathcal{D}_r$) and forgetting ($\mathcal{D}_f$) performance requires an appropriate $\alpha$. After defining calibration targets $Cal(\mathcal{D}_r)$ and $Cal(\mathcal{D}_f)$, we can monitor accuracies $Acc_u(\mathcal{D}_r)$ and $Acc_u(\mathcal{D}_f)$ during unlearning. The calibration target $Cal$ defines the reference accuracy we expect the unlearning model to achieve on forgetting and retaining datasets during training. And **the gap between observed accuracies ($Acc_u(\mathcal{D}_r)$, $Acc_u(\mathcal{D}_f)$) and their corresponding calibration targets ($Cal(\mathcal{D}_r)$, $Cal(\mathcal{D}_f)$) guides the adaptive adjustment of $\alpha$.**

---

**Calibration Principles behind Adaptive $\beta$** [1]

① $Acc_u(\mathcal{D}_r)$: $Cal(\mathcal{D}_r) = Acc_o(\mathcal{D}_{train})$. The unlearning model's retaining accuracy ($Acc_u(\mathcal{D}_r)$) should be preserved as the original model's training accuracy ($Acc_o(\mathcal{D}_{train})$).

② $Acc_u(\mathcal{D}_f)$: $Cal(\mathcal{D}_f) = Acc_o(\mathcal{D}_v)$ for random data forgetting, $Cal(\mathcal{D}_f) = 0$ for class-wise forgetting. Since $\mathcal{D}_f$ should be unlearned as if it were never trained, unlearning model's forgetting accuracy ($Acc_u(\mathcal{D}_f)$) should have the same level as the original model's validation accuracy ($Acc_o(\mathcal{D}_v)$) in random data forgetting, and should be 0 in class-wise forgetting.

---

Guided by these rationales, the three calibration conditions are listed as follows:

- **Condition ❶.** When $Acc_u(\mathcal{D}_f) \leq Cal(\mathcal{D}_f)$, it indicates that the model has successfully forgotten $\mathcal{D}_f$ or even over-forgotten it. In this case, we set $\beta = 0$ to prevent further forgetting.

- **Condition ❷.** If $Acc_u(\mathcal{D}_f) > Cal(\mathcal{D}_f)$ and the performance degradation on the $\mathcal{D}_r$ is more severe than that on $\mathcal{D}_f$, a mild forgetting adjustment can be set to $\beta = 0.1$.

---

[1]The $Acc_o(\mathcal{D}_{train})$ and $Acc_o(\mathcal{D}_v)$ are constants as they are recorded during the training of $\theta_o$. Since the unlearning process is controlled by the data owner, it is reasonable to assume access to a small validation set. In our experiments, we split the original test set into 10% for $\mathcal{D}_v$ and 90% for $\mathcal{D}_t$.

- **Condition ❸.** Otherwise, we apply a stronger forgetting adjustment with $\beta = 0.5$.

The adaptive adjustment of $\beta$ is formulated as follows,

$$\beta = \begin{cases} 0, & Acc_u(\mathcal{D}_f) \leq Cal(\mathcal{D}_f), \ (❶) \\ 0.1, & Acc_u(\mathcal{D}_f) > Cal(\mathcal{D}_f) \text{ and} \\ & \frac{Acc_u(\mathcal{D}_f) - Cal(\mathcal{D}_f)}{Cal(\mathcal{D}_f)} < \frac{Acc_u(\mathcal{D}_r) - Cal(\mathcal{D}_r)}{Cal(\mathcal{D}_r)}, \ (❷) \\ 0.5, & otherwise. \ (❸) \end{cases}$$

(7)

Unlike fixed hyperparameter tuning, our adaptive $\beta$ strategy updates dynamically at every batch within each training epoch. This adaptive adjustment ensures that the unlearning process remains responsive to the pathway's evolving state, striking a balance between effective forgetting and retaining. Furthermore, another notable benefit of the adaptive $\beta$ adjustment is that the MCU becomes less sensitive to the parameters $k$ and $k_r$ and scarce retaining data (see Figures 11-14 in Appendix D for experimental results).

## 3.5. Optimal Model and Effective Region

As illustrated in Figure 1c, an important step after the pathway searching is to identify the optimal unlearning model and effective unlearning region along the pathway. Following the calibration principles introduced in Section 3.4, we utilize constants $Cal(\mathcal{D}_r)$ and $Cal(\mathcal{D}_f)$ as accuracy reference values for computing calibration gaps. These gaps quantify the deviation between unlearning models and the desired behavior, guiding both optimal model and effective region identification.

The model selection process along the Bézier curve is conducted during inference, and thus incurs negligible computational overhead. **To efficiently locate the optimal model**, we first evaluate models at $t = 0.75$ and $t = 1$, and then perform a cubic interpolation of their accuracy values to estimate the $t$ value with minimal gap as the optimal model point. This heuristic is motivated by our empirical observation that the optimal model along the pathway always lies within the interval $t \in [0.75, 1]$. This strategy avoids exhaustive sampling across the entire pathway. **For identifying the effective region**, we uniformly sample 20 points along $t \in [0, 1]$ and fit a cubic interpolation curve. Any point on the continuous curve whose gap is smaller than that of the pre-unlearning model (at $t = 1$) as part of the effective region.

## 4. Experiments

### 4.1. Experiment Setups

**Datasets and Models.** We focus on image classification tasks under random data and class-wise forgetting scenarios, using 3 datasets (**CIFAR-10**, **ImageNet-100**,

*Table 1.* Overall performance of MU methods under $10\%$ **random data forgetting** scenario. The results are presented in the format $a \pm b$, with $a$ as the mean and $b$ as the standard deviation from 5 independent trials. The performance gap relative to RT method is represented in (•). The Avg. Gap is derived by averaging gaps across accuracy metrics, UA, RA, TA and MIA. Smaller gaps reflect closer alignment with the RT model's performance. Note RTE is reported in minutes and UA equals $1-$ accuracy of $\mathcal{D}_f$.

| Methods | UA ↓ | RA ↓ | TA ↓ | MIA ↓ | Avg. Gap ↓ | RTE ↓ |
|---|---|---|---|---|---|---|
| **CIFAR-10 with PreResNet-110** | | | | | | |
| RT | $10.54_{\pm0.34}(0.00)$ | $99.98_{\pm0.01}(0.00)$ | $89.59_{\pm0.22}(0.00)$ | $18.41_{\pm0.52}(0.00)$ | $0.00$ | $105.70$ |
| FT | $0.42_{\pm0.12}(10.12)$ | $99.93_{\pm0.01}(0.05)$ | $90.99_{\pm0.13}(1.40)$ | $3.71_{\pm0.25}(14.70)$ | $6.57$ | $5.31$ |
| RL | $4.14_{\pm0.20}(6.40)$ | $99.69_{\pm0.02}(0.29)$ | $90.16_{\pm0.09}(0.57)$ | $21.93_{\pm0.66}(3.52)$ | $2.70$ | $6.21$ |
| GA | $0.06_{\pm0.00}(10.48)$ | $99.97_{\pm0.00}(0.01)$ | $90.89_{\pm0.01}(1.30)$ | $0.98_{\pm0.01}(17.43)$ | $7.31$ | $0.38$ |
| NegGrad+ | $7.03_{\pm0.32}(3.51)$ | $98.63_{\pm0.19}(1.35)$ | $89.26_{\pm0.23}(0.33)$ | $11.71_{\pm0.38}(6.70)$ | $2.97$ | $2.96$ |
| SFRon | $14.07_{\pm0.19}(3.53)$ | $92.86_{\pm0.16}(7.12)$ | $85.74_{\pm0.12}(3.85)$ | $14.32_{\pm0.44}(4.09)$ | $4.65$ | $2.04$ |
| SalUn | $6.67_{\pm0.26}(3.87)$ | $97.87_{\pm0.14}(2.11)$ | $90.54_{\pm0.19}(0.95)$ | $35.45_{\pm0.57}(17.04)$ | $5.99$ | $6.38$ |
| NegTV | $2.36_{\pm1.12}(8.18)$ | $99.08_{\pm0.60}(0.90)$ | $88.53_{\pm0.88}(1.06)$ | $4.14_{\pm0.29}(14.27)$ | $6.10$ | $0.70$ |
| MCU | $9.52_{\pm0.04}(1.02)$ | $98.97_{\pm0.01}(1.01)$ | $89.00_{\pm0.03}(0.59)$ | $16.33_{\pm0.93}(2.08)$ | $1.18$ | $6.80$ |
| $MCU_\beta$ | $10.29_{\pm0.24}(0.25)$ | $98.69_{\pm0.04}(1.29)$ | $89.11_{\pm0.13}(0.48)$ | $16.45_{\pm0.89}(1.96)$ | $\mathbf{1.00}$ | $6.82$ |
| **ImageNet-100 with ViT** | | | | | | |
| RT | $11.63_{\pm0.23}(0.00)$ | $91.93_{\pm0.01}(0.00)$ | $87.83_{\pm0.01}(0.00)$ | $13.77_{\pm0.42}(0.00)$ | $0.00$ | $525.72$ |
| FT | $8.62_{\pm0.01}(3.01)$ | $92.21_{\pm0.07}(0.28)$ | $87.74_{\pm0.18}(0.09)$ | $10.88_{\pm0.43}(2.89)$ | $1.57$ | $82.23$ |
| RL | $9.53_{\pm0.15}(2.10)$ | $92.06_{\pm0.02}(0.13)$ | $87.82_{\pm0.10}(0.01)$ | $24.32_{\pm0.35}(10.55)$ | $3.20$ | $205.73$ |
| GA | $8.96_{\pm0.89}(2.67)$ | $91.15_{\pm0.58}(0.78)$ | $87.53_{\pm0.37}(0.30)$ | $10.50_{\pm0.07}(3.27)$ | $1.76$ | $6.71$ |
| NegGrad+ | $13.15_{\pm0.10}(1.52)$ | $91.71_{\pm0.03}(0.22)$ | $87.37_{\pm0.07}(0.46)$ | $16.21_{\pm0.30}(2.44)$ | $1.16$ | $63.93$ |
| SFRon | $27.28_{\pm0.37}(15.65)$ | $78.56_{\pm1.51}(13.37)$ | $77.86_{\pm1.24}(9.97)$ | $61.29_{\pm0.62}(47.52)$ | $21.63$ | $88.37$ |
| SalUn | $9.38_{\pm0.13}(2.25)$ | $91.94_{\pm0.03}(0.00)$ | $87.73_{\pm0.13}(0.10)$ | $24.29_{\pm1.00}(10.52)$ | $3.22$ | $170.34$ |
| NegTV | $10.17_{\pm0.10}(1.46)$ | $91.33_{\pm0.09}(0.60)$ | $87.24_{\pm0.04}(0.59)$ | $12.25_{\pm0.21}(1.52)$ | $1.04$ | $11.02$ |
| MCU | $11.44_{\pm0.04}(0.19)$ | $92.02_{\pm0.02}(0.09)$ | $87.62_{\pm0.08}(0.21)$ | $16.33_{\pm0.18}(2.56)$ | $0.76$ | $103.47$ |
| $MCU_\beta$ | $11.63_{\pm0.08}(0.00)$ | $91.92_{\pm0.10}(0.01)$ | $87.70_{\pm0.11}(0.13)$ | $16.21_{\pm0.31}(2.44)$ | $\mathbf{0.65}$ | $103.52$ |

*Table 2.* Unlearning performance of MU methods for **class-wise forgetting** in **ImageNet-100** with **ViT**. The table adopts the same format as Table 1. $UA_{test}$ is the unlearning accuracy on test-forgetting data $\mathcal{D}_{tf}$.

| Methods | UA ↓ | $UA_{test}$ ↓ | RA ↓ | TA ↓ | MIA ↓ | Avg. Gap ↓ | RTE ↓ |
|---|---|---|---|---|---|---|---|
| RT | $100.00_{\pm0.00}(0.00)$ | $100.00_{\pm0.00}(0.00)$ | $92.01_{\pm0.08}(0.00)$ | $88.17_{\pm0.11}(0.00)$ | $100.00_{\pm0.00}(0.00)$ | $0.00$ | $606.93$ |
| FT | $80.69_{\pm2.62}(19.31)$ | $83.00_{\pm1.00}(17.00)$ | $92.33_{\pm0.04}(0.32)$ | $87.82_{\pm0.04}(0.35)$ | $83.27_{\pm3.81}(16.73)$ | $10.74$ | $100.68$ |
| RL | $96.15_{\pm0.46}(3.85)$ | $100.00_{\pm0.00}(0.00)$ | $92.21_{\pm0.07}(0.20)$ | $88.10_{\pm0.04}(0.07)$ | $100.00_{\pm0.00}(0.00)$ | $0.82$ | $200.23$ |
| GA | $100.00_{\pm0.00}(0.00)$ | $100.00_{\pm0.00}(0.00)$ | $81.42_{\pm1.99}(10.59)$ | $78.11_{\pm2.03}(10.06)$ | $100.00_{\pm0.00}(0.00)$ | $4.13$ | $0.76$ |
| NegGrad+ | $97.46_{\pm1.34}(2.54)$ | $99.00_{\pm1.00}(1.00)$ | $92.17_{\pm0.03}(0.16)$ | $87.90_{\pm0.06}(0.27)$ | $96.58_{\pm0.27}(3.42)$ | $1.48$ | $69.14$ |
| SFRon | $100.00_{\pm0.00}(0.00)$ | $100.00_{\pm0.00}(0.00)$ | $81.38_{\pm0.11}(10.63)$ | $80.97_{\pm0.18}(7.20)$ | $100.00_{\pm0.00}(0.00)$ | $3.57$ | $87.96$ |
| SalUn | $95.35_{\pm0.88}(4.65)$ | $100.00_{\pm0.00}(0.00)$ | $92.06_{\pm0.09}(0.05)$ | $88.01_{\pm0.01}(0.16)$ | $100.00_{\pm0.00}(0.00)$ | $0.97$ | $174.67$ |
| NegTV | $97.85_{\pm0.15}(2.15)$ | $100.00_{\pm0.00}(0.00)$ | $91.39_{\pm0.02}(0.62)$ | $87.60_{\pm0.02}(0.57)$ | $99.15_{\pm0.00}(0.85)$ | $0.84$ | $1.24$ |
| MCU | $100.00_{\pm0.00}(0.00)$ | $100.00_{\pm0.00}(0.00)$ | $92.32_{\pm0.03}(0.21)$ | $87.92_{\pm0.11}(0.25)$ | $100.00_{\pm0.00}(0.00)$ | $0.09$ | $105.49$ |
| $MCU_\beta$ | $100.00_{\pm0.00}(0.00)$ | $100.00_{\pm0.00}(0.00)$ | $92.18_{\pm0.05}(0.17)$ | $88.00_{\pm0.09}(0.17)$ | $100.00_{\pm0.00}(0.00)$ | $\mathbf{0.07}$ | $98.12$ |

**Tiny-ImageNet**) and 3 architectures (**PreResNet-110**, **ViT**, **VGG-16-BN**). See Appendix C for forgetting scenario details.

**Baselines and Metrics.** We compare our framework against 8 methods: Retrain (**RT**), Finetune (**FT**) (Warnecke et al., 2021), Random Label (**RL**) (Graves et al., 2021), Gradient Ascent (**GA**) (Thudi et al., 2022), **NegGrad+** (Kurmanji et al., 2024a), **SFRon** (Huang et al., 2025), **SalUn** (Fan et al., 2023), **NegTV** (Ilharco et al., 2022). See Appendix C for detailed introduction of these baselines. We denote our framework with fixed $\beta$ as **MCU**[2] and with adaptive $\beta$ as **MCU$_\beta$**. Unless otherwise stated, the pre-unlearning model in MCUs is NegGrad+. We evaluate all methods across five metrics: **UA** (Unlearning Accuracy, $1-$ accuracy of forgetting data $\mathcal{D}_f$), **RA** (Retaining Accuracy, accuracy on retaining data $\mathcal{D}_r$), **TA** (Test Accuracy, accuracy on test data $\mathcal{D}_t$), **MIA** (Membership Inference Attack, see Appendix C for the details), and **RTE** (Running

Time Efficiency). Except for RTE, all metrics are evaluated based on their proximity to the RT baseline, with smaller average gap indicating better unlearning performance (denoted as Avg. Gap in our result tables).

### 4.2. Experiment Results

**Overall Performance.** We evaluate the performance of MU baselines and our framework MCU and MCU$_\beta$. Table 1 presents results for $10\%$ random data forgetting across 2 datasets and architectures, while Table 2 reports results for class-wise forgetting on ImageNet-100 dataset with ViT. Additional results on other datasets, architectures, and unlearning scenarios are included in Tables 5-9 in Appendix D. The Avg. Gap presents the mean performance gap across UA, RA, TA, and MIA.

Under comprehensive metrics, both MCU and MCU$_\beta$ consistently exhibit the top two overall performances under both random data forgetting and class-wise forgetting. Notably, in the class-wise forgetting scenario, MCU$_\beta$ performs nearly

---

[2]The best results achieved through hyperparameter $\beta$ search.

*Table 3.* Unlearning performance of different pre-unlearning models in $MCU_\beta$. The results demonstrate that applying our $MCU_\beta$ framework to any unlearning method can significantly enhance unlearning performance.

| Methods | UA↓ | RA↓ | TA↓ | MIA↓ | Avg. Gap↓ | RTE↓ |
|---|---|---|---|---|---|---|
| RT | $10.54_{\pm0.34}(0.00)$ | $99.98_{\pm0.01}(0.00)$ | $89.59_{\pm0.22}(0.00)$ | $18.41_{\pm0.52}(0.00)$ | 0.00 | 105.70 |
| FT | $0.42_{\pm0.12}(10.12)$ | $99.93_{\pm0.01}(0.05)$ | $90.99_{\pm0.13}(1.40)$ | $3.71_{\pm0.25}(14.70)$ | 6.57 | 5.31 |
| $MCU_\beta$-FT | $5.62_{\pm0.07}(4.92)$ | $99.12_{\pm0.02}(0.86)$ | $89.59_{\pm0.10}(0.00)$ | $10.68_{\pm0.62}(7.73)$ | **3.37** | 9.76 |
| RL | $4.14_{\pm0.20}(6.40)$ | $99.69_{\pm0.02}(0.29)$ | $90.16_{\pm0.09}(0.57)$ | $21.93_{\pm0.66}(3.52)$ | 2.70 | 6.21 |
| $MCU_\beta$-RL | $10.54_{\pm0.02}(0.00)$ | $98.60_{\pm0.11}(1.38)$ | $89.21_{\pm0.08}(0.38)$ | $22.48_{\pm0.47}(4.07)$ | **1.46** | 12.24 |
| GA | $0.06_{\pm0.00}(10.48)$ | $99.97_{\pm0.00}(0.01)$ | $90.89_{\pm0.01}(1.30)$ | $0.98_{\pm0.11}(17.43)$ | 7.31 | 0.38 |
| $MCU_\beta$-GA | $3.84_{\pm0.01}(6.70)$ | $98.80_{\pm0.05}(1.18)$ | $88.86_{\pm0.33}(0.73)$ | $13.22_{\pm0.37}(5.19)$ | **3.45** | 5.03 |
| NegGrad+ | $7.03_{\pm0.32}(3.51)$ | $98.63_{\pm0.23}(1.35)$ | $89.26_{\pm0.23}(0.33)$ | $11.71_{\pm0.38}(6.70)$ | 2.97 | 2.96 |
| $MCU_\beta$-NegGrad+ | $10.29_{\pm0.24}(0.25)$ | $98.69_{\pm0.04}(1.29)$ | $89.11_{\pm0.13}(0.48)$ | $16.45_{\pm0.89}(1.96)$ | **1.00** | 6.82 |
| SFRon | $14.07_{\pm0.19}(3.53)$ | $92.86_{\pm0.16}(7.12)$ | $85.74_{\pm0.12}(3.85)$ | $14.32_{\pm0.44}(4.09)$ | 4.65 | 2.04 |
| $MCU_\beta$-SFRon | $8.82_{\pm1.71}(1.72)$ | $96.19_{\pm1.05}(3.79)$ | $88.11_{\pm0.61}(1.48)$ | $15.12_{\pm0.14}(3.29)$ | **2.57** | 6.93 |
| SalUn | $6.67_{\pm0.26}(3.87)$ | $97.87_{\pm0.14}(2.11)$ | $90.54_{\pm0.19}(0.95)$ | $35.45_{\pm0.57}(17.04)$ | 5.99 | 6.38 |
| $MCU_\beta$-SalUn | $10.49_{\pm0.07}(0.05)$ | $97.55_{\pm0.10}(2.43)$ | $89.21_{\pm0.23}(0.38)$ | $30.30_{\pm1.17}(11.89)$ | **3.68** | 11.35 |
| NegTV | $2.36_{\pm1.12}(8.18)$ | $99.08_{\pm0.60}(0.90)$ | $88.53_{\pm0.88}(1.06)$ | $4.14_{\pm0.29}(14.27)$ | 6.10 | 0.70 |
| $MCU_\beta$-NegTV | $8.11_{\pm0.60}(2.43)$ | $98.01_{\pm0.32}(1.97)$ | $87.74_{\pm0.33}(1.85)$ | $11.48_{\pm0.18}(6.93)$ | **3.30** | 5.67 |

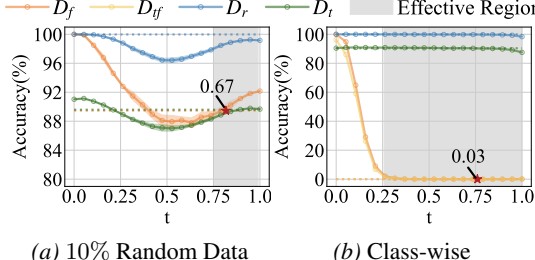

*(a)* 10% Random Data     *(b)* Class-wise

*Figure 3.* Effective unlearning region on $MCU_\beta$. The marker ★ highlights the position with the minimum average gap from RT, with the accompanying numerical value indicating the exact average accuracy gap of $\mathcal{D}_f$, $\mathcal{D}_r$ and $\mathcal{D}_t$ (and $\mathcal{D}_{tf}$ for class-wise forgetting). The dotted line represents the RT method's accuracy, serving as a reference. The shaded gray area denotes the effective unlearning region, where models achieve better unlearning performance than the $\theta_p$.

on par with the RT method. Additionally, $MCU_\beta$ outperforms MCU, validating our adaptive $\beta$ strategy, which both simplifies training process and enhances the effectiveness of MCU framework.

The results highlight the superiority of nonlinear unlearning over the linear method NegTV, especially in the class-wise forgetting scenario in Tables 7 and 8 in the Appendix D. In the class-wise forgetting scenario, we attempted to optimize NegTV by extensively tuning its scaling hyperparameter, but encountered a persistent dilemma: NegTV either resulted in under-forgetting or over-forgetting. This stark trade-off highlights the inherent challenge of weight entanglement in linear approaches, which struggle to achieve the balance required for effective class-wise unlearning.

As a strong baseline, SalUn and RL is generally second only to MCUs and performs especially well in the class-wise forgetting scenario. However, SalUn and RL tend to exhibit overly strong resistance to MIA, often deviating significantly from RT in terms of membership privacy. While higher MIA efficacy is typically desirable for privacy, in the context of MU, the goal is to align with the RT baseline rather

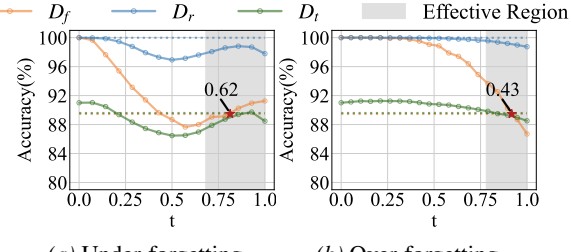

*(a)* Under-forgetting     *(b)* Over-forgetting

*Figure 4.* Effectiveness of $MCU_\beta$ across both under-forgetting and over-forgetting pre-unlearning model $\theta_p$.

than excessively suppress MIA scores. Excessive deviation from RT could indicate a shift in model behavior that may introduce unintended privacy risks.

**Fairness of Model Selection.** To ensure fair comparison, we also applied our optimal model selection strategy (Section 3.5) to single model baselines. Specifically, we allow post-hoc selection over a baseline's training checkpoints using the same calibration criteria used in MCU. **The results in Figure 15 (Appendix D) demonstrate that MCU's advantage fundamentally stems from nonlinear interpolation in continuous parameter space, rather than merely from the post-hoc selection strategy.**

**Effective Unlearning Region.** Figure 3 shows visualization results of $MCU_\beta$ on CIFAR-10 with PreResNet-110 under both 10% random data forgetting and class-wise forgetting scenarios. The results demonstrate that $MCU_\beta$ not only identifies a single effective unlearning model but also discovers a substantial region along the Bézier pathway where multiple models in this pathway exhibit effective unlearning. Within this effective unlearning region, models achieve superior unlearning performance compared to the pre-unlearning model ($t = 1.0$). Moreover, $MCU_\beta$ provides greater flexibility since different effective unlearning models can be selected based on task-specific requirements. For instance, in Figure 3a, models to the right of marker ★ preserve better predictive performance, while those to the

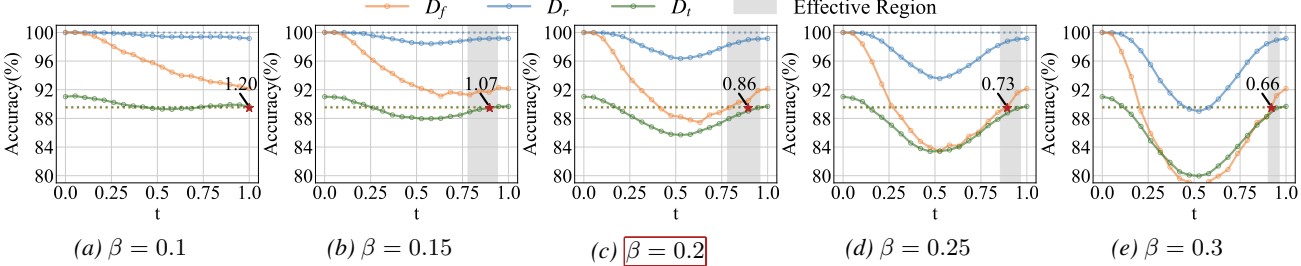

*Figure 5.* Ablation study for $\beta$ on MCU. Overall, increasing $\beta$ effectively enhances the unlearning effect but damages retaining predictive performance, while decreasing $\beta$ weakens the ability of the pathway to forget data.

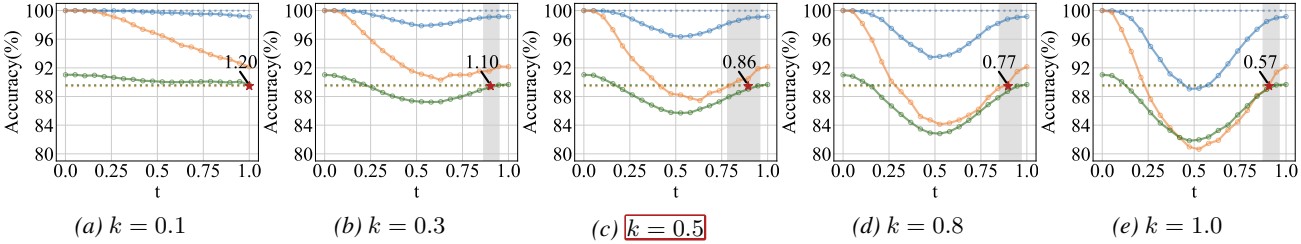

*Figure 6.* Ablation study for $k$ on MCU. As $k$ increases, the average accuracy gap decreases, but the effective region also shrinks.

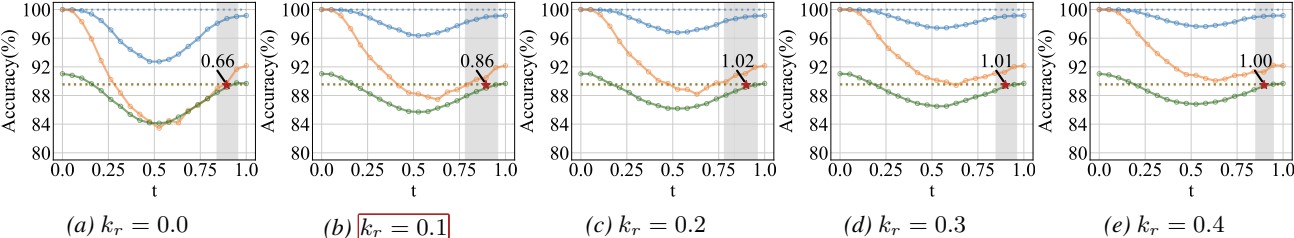

*Figure 7.* Ablation study for $k_r$ on MCU. When $k_r = 0$, we preserve and update all parameters important to retaining data, leading to a noticeable drop in $\mathcal{D}_r$ accuracy during the unlearning process.

left demonstrate stronger forgetting efficacy.

**Effectiveness in Different Pre-unlearning Models.** In this experiment, we integrate various MU methods as pre-unlearning models into our MCU$_\beta$ framework under 10% random data forgetting scenario on CIFAR-10 with PreResNet-110. Table 3 compares the performance of these methods before and after incorporating the MCU$_\beta$. The results demonstrate that MCU$_\beta$ consistently enhances the performance of all MU methods. On average, the Avg. Gap across all methods is reduced by 48.99%, with particularly notable improvements in the UA metric. We further evaluate robustness using intentionally degraded pre-unlearning models in Figure 12 (Appendix D), demonstrating that MCU remains effective in improving unlearning performance even under adverse conditions. These results highlight the robustness of MCU across different $\theta_p$.

**Effectiveness across Under-forgetting and Over-forgetting Pre-unlearning Models.** To further demonstrate the versatility of MCU$_\beta$, we evaluate its ability to handle both under-forgetting and over-forgetting scenarios of pre-unlearning models with the same data and architecture setting as Figure 3a. While Figure 4a

shows the under-forgetting case where RL is trained for 15 epochs, we intentionally over-trained RL for 20 epochs as an over-forgetting pre-unlearning model in Figure 4b. As shown in Figure 4, MCU$_\beta$-RL consistently enhances RL in both scenarios. Specifically, it reduces the average gap across $D_f$, $D_r$, $D_t$ to 0.62 in the under-forgetting scenario and 0.43 in the over-forgetting scenario. These results highlight MCU$_\beta$'s adaptability across different pre-unlearning conditions. This is attributed to the adaptive unlearning penalty coefficient $\beta$, with the calibration condition ❶ handling over-forgetting and conditions ❷ and ❸ handling under-forgetting.

**Ablation Study.** To better understand the role of hyper-parameters, $\beta$, $k$, and $k_r$, we conduct an ablation study on CIFAR-10 with PreResNet-110 under 10% random data forgetting. Figures 5-7 follow the same format as Figure 3, with red-framed sub-captions indicating our default settings ($\beta = 0.2$, $k = 0.5$, $k_r = 0.1$). For each ablation experiment, we vary one parameter while keeping the others at default.

A higher $\beta$ value leads to a smaller average accuracy gap in Figure 5. Notably, when $\beta = 0.3$, the average gap is only 0.66. However, increasing $\beta$ also results in a reduced effec-

*Table 4.* Unlearning performance of the Polychain pathway for **10% random data forgetting** in **CIFAR-10** with **PreResNet-110**. Polychain-$k$ corresponds to a Polychain pathway parameterized with $c$ control points, where $c \in \{1, 2, 3\}$.

| Methods | UA $\downarrow$ | RA $\downarrow$ | TA $\downarrow$ | MIA $\downarrow$ | Avg. Gap $\downarrow$ | RTE $\downarrow$ |
|---|---|---|---|---|---|---|
| RT | $10.54_{\pm 0.34}(0.00)$ | $99.98_{\pm 0.01}(0.00)$ | $89.59_{\pm 0.22}(0.00)$ | $18.41_{\pm 0.52}(0.00)$ | $0.00$ | $105.70$ |
| Polychain-1 | $8.90_{\pm 0.89}(1.64)$ | $98.81_{\pm 0.03}(1.17)$ | $89.58_{\pm 0.04}(0.01)$ | $17.13_{\pm 0.72}(1.28)$ | $1.03$ | $6.76$ |
| Polychain-2 | $7.85_{\pm 1.11}(2.69)$ | $98.92_{\pm 0.05}(1.06)$ | $89.59_{\pm 0.07}(0.00)$ | $16.01_{\pm 0.62}(2.40)$ | $1.53$ | $6.86$ |
| Polychain-3 | $8.68_{\pm 1.24}(1.86)$ | $98.43_{\pm 0.10}(1.55)$ | $89.58_{\pm 0.09}(0.01)$ | $16.72_{\pm 1.32}(1.69)$ | $1.28$ | $6.90$ |
| Bézier | $10.29_{\pm 0.24}(0.25)$ | $98.69_{\pm 0.04}(1.29)$ | $89.11_{\pm 0.13}(0.48)$ | $16.45_{\pm 0.89}(1.96)$ | **1.00** | $6.82$ |

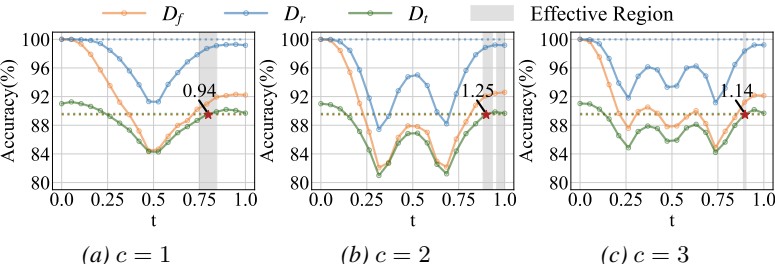

*Figure 8.* Polychain pathway with different number of control points for 10% random data forgetting in CIFAR10 with PreResNet-110. $c$ denotes Polychain with $c$ control points ($c = 1, 2, 3$).

tive region. This suggests that while a larger $\beta$ improves forgetting, it leads to a degradation in model utility. Clearly, $\beta = 0.2$ offers the best balance between average accuracy gap and effective region. Nonetheless, choosing a larger $\beta$ can still be a viable and wise option when minimizing the accuracy gap is the primary objective, and the effective region is of secondary importance.

Similarly, we analyze the impact of $k$ and $k_r$, in our mask strategy. A larger $k$ allows more parameters retained for training, which significantly reduces the accuracy on $\mathcal{D}_f$, $\mathcal{D}_r$, and $\mathcal{D}_t$, especially $\mathcal{D}_f$ (orange lines in Figure 6). As for $k_r$, increasing $k_r$ results in the removal of essential parameters related to $\mathcal{D}_r$, thereby effectively preserving the accuracy on $\mathcal{D}_r$ (blue lines in Figure 7). In our experiments, we set $k = 0.5$ and $k_r = 0.1$ as default values, as they provide a good balance between enhancing forgetting quality and maintaining predictive performance.

**MCU$_\beta$ with Polychain Unlearning Pathway.** Here, we further explore MCU$_\beta$ framework on Polychain. The Polychain pathway with one control point in parameter space is defined as follows,

$$\phi_{\boldsymbol{\theta}_c}(t) = \begin{cases} 2\left(t\boldsymbol{\theta}_c + (0.5 - t)\boldsymbol{\theta}_o\right), & 0 \leq t \leq 0.5, \\ 2\left((t - 0.5)\boldsymbol{\theta}_p + (1 - t)\boldsymbol{\theta}_c\right), & 0.5 < t \leq 1. \end{cases}$$
(8)

$\boldsymbol{\theta}_c$ is the control model, and $t$ represents a scalar interpolation coefficient that controls the position along the pathway connecting two end models in the high-dimensional parameter space.

The Polychain results on CIFAR-10 with PreResNet-110

under 10% random data forgetting scenario are shown in Table 4 and Figure 8. Table 4 shows that the Bézier curve achieves better unlearning performance than the Polychain, with a lower Avg. Gap. Nevertheless, the unlearning performance of our MCU$_\beta$ framework, whether using the Polychain or Bézier pathway, still surpasses all baseline methods reported in Table 1. Moreover, as illustrated in Figure 8, the accuracy trajectory of the Polychain is more irregular than that of the Bézier curve, exhibiting noticeable turning points. Moreover, increasing the number of control points in the Polychain does not lead to improvements in its unlearning pathway. The Bézier curve is generally superior because it offers smoother and more flexible paths for connecting models. The Bézier curves can define a continuous, differentiable trajectory, making them well-suited for efficient optimization and avoiding sharp transitions that can lead to instability during unlearning. These advantages have led many prior works to only adopt Bézier curves or surfaces for their studies (Li et al., 2025; Ren et al.; Zhao et al., 2020). Therefore, we adopt the Bézier curve as our default setting.

## 5. Conclusion

In this work, we propose a novel framework MCU, leveraging mode connectivity to search nonlinear pathway in parameter space for effective unlearning. Unlike traditional MU methods that identify only a single unlearning model, MCU uncovers a spectrum of unlearning models along the pathway and is free from empirical hyperparameter tuning. As a plug-and-play framework, MCU seamlessly integrates with existing MU methods and consistently improves their unlearning efficacy.

## Impact Statement

This paper presents work whose goal is to advance the field of Machine Learning. There are many potential societal consequences of our work, none of which we feel must be specifically highlighted here.

## Acknowledgments

This work was supported in part by the National Science Foundation under grants IIS-2246157, FMitF-2319243, and the Department of Energy under grant DE-CR0000042. The project was also supported by computational resources provided by the NSF ACCESS and Argonne National Lab.

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

# Appendix

## A. Weight Entanglement in Linear MU Method

In this section, we analyze the weight entanglement issue that arises in linear MU methods, i.e., task arithmetic (Ilharco et al., 2022; Iurada et al.; Ortiz-Jimenez et al., 2024). Let $f : \mathcal{X} \times \Theta \to \mathcal{Y}$ be a neural network that takes input $\boldsymbol{x} \in \mathcal{X}$ and is parameterized by $\boldsymbol{\theta} \in \Theta$. We assume $\mathcal{X} \subseteq \mathbb{R}^d$, $\Theta \subseteq \mathbb{R}^m$, and $\mathcal{Y} \subseteq \mathbb{R}^c$. Given the original model parameters $\boldsymbol{\theta}_o \in \mathbb{R}^m$, a fine-tuned model with parameters $\boldsymbol{\theta}_{ft}^f$ is trained on the forgetting dataset $\mathcal{D}_f$.

The unlearning task vector is defined as the difference between the fine-tuned and original model parameters, i.e., $\boldsymbol{\tau}_f = \boldsymbol{\theta}_{ft}^f - \boldsymbol{\theta}_o$ where $\boldsymbol{\theta}_{ft}^f$ is fine-tuned on forgetting data $\mathcal{D}_f$ based on $\boldsymbol{\theta}_o$. By task arithmetic, it is easy to manipulate the output behavior of the model by adding or subtracting task vectors. Thus, in our unlearning scenario, the unlearning model can be defined with the negation task vector as:

$$f(\boldsymbol{x}; \boldsymbol{\theta}_u) = f(\boldsymbol{x}; \boldsymbol{\theta}_o - \alpha\boldsymbol{\tau}_f) = f\left(\boldsymbol{x}; \boldsymbol{\theta}_o - \alpha(\boldsymbol{\theta}_{ft}^f - \boldsymbol{\theta}_o)\right), \tag{9}$$

where $\alpha$ is a coefficient that controls forgetting level. This formulation implicitly requires that subtracting the task vector $\boldsymbol{\tau}_f$ does not affect the model's predictions on inputs outside the forgetting data $\mathcal{D}_f$. In other words, $\boldsymbol{\tau}_f$ should not encode any information about data outside $\mathcal{D}_f$, i.e., retaining data $\mathcal{D}_r$. Therefore, the condition for this equation to hold can be formalized as:

$$f\left(\boldsymbol{x}; \boldsymbol{\theta}_o - \alpha\boldsymbol{\tau}_f\right) = \begin{cases} f(\boldsymbol{x}; \boldsymbol{\theta}_o), & \boldsymbol{x} \in \mathcal{D}_r \\ f(\boldsymbol{x}; \boldsymbol{\theta}_o - \alpha\boldsymbol{\tau}_f), & \boldsymbol{x} \in \mathcal{D}_f. \end{cases} \tag{10}$$

This condition requires that the task vector $\boldsymbol{\tau}_f$ in Eq. 9 only influences the model on the forgetting dataset, leaving the performance on retaining data $\mathcal{D}_r$ unaffected. However, task vectors obtained via simple fine-tuning on $\mathcal{D}_f$ do not guarantee this condition, which faces a weight entanglement issue.

To address this, the model must exhibit a form of weight disentanglement. Ideally, the model should behave as a composition of spatially localized components, each responsible for a specific data domain. For our unlearning case, this means the function should decompose as:

$$\begin{aligned} & f(\boldsymbol{x}; \boldsymbol{\theta}_o - \alpha\boldsymbol{\tau}_f) \\ & = f(\boldsymbol{x}; \boldsymbol{\theta}_o)\mathbb{1}(\boldsymbol{x} \in \mathcal{D}_r) + f(\boldsymbol{x}; \boldsymbol{\theta}_o - \alpha\boldsymbol{\tau}_f)\mathbb{1}(\boldsymbol{x} \in \mathcal{D}_f) \\ & = g_o(\boldsymbol{x}) + g_f(\boldsymbol{x}; -\alpha\boldsymbol{\tau}_f), \end{aligned} \tag{11}$$

The term $g_o(\boldsymbol{x}) := f(\boldsymbol{x}; \boldsymbol{\theta}_o) \cdot \mathbb{1}(\boldsymbol{x} \in \mathcal{D}_r)$ denotes spatially localized components for retaining data domain, and $g_o(\boldsymbol{x}) = 0$ for $\boldsymbol{x} \in \mathcal{D}_f$. The term $g_f(\boldsymbol{x}; -\alpha\boldsymbol{\tau}_f) := f(\boldsymbol{x}; \boldsymbol{\theta}_o - \alpha\boldsymbol{\tau}_f) \cdot \mathbb{1}(\boldsymbol{x} \in \mathcal{D}_f)$ captures the influence of the unlearning task vector, localized within the forgetting data domain, and $g_f(\boldsymbol{x}; \alpha\boldsymbol{\tau}_f) = 0$ for $\boldsymbol{x} \in \mathcal{D}_r$. This decomposition encapsulates the principle that only data within $\mathcal{D}_f$ should be influenced by $\boldsymbol{\tau}_f$.

To make this decomposition tractable, linearizing the network around $\boldsymbol{\theta}_o$ via a first-order Taylor expansion is attempted to realize it by :

$$\begin{aligned} & f(\boldsymbol{x}; \boldsymbol{\theta}_o - \alpha\boldsymbol{\tau}_f) \\ & \approx f_{\text{lin}}(\boldsymbol{x}; \boldsymbol{\theta}_o - \alpha\boldsymbol{\tau}_f) = f(\boldsymbol{x}; \boldsymbol{\theta}_o) - \alpha\boldsymbol{\tau}_f^\top \nabla_{\boldsymbol{\theta}} f(\boldsymbol{x}; \boldsymbol{\theta}_o). \end{aligned} \tag{12}$$

This linearized model expresses the output as a combination of the original prediction and a perturbation determined by the gradient of $f$ at $\boldsymbol{\theta}_o$.

While this form resembles the disentangled decomposition in Eq. 11, this resemblance is superficial. The disentanglement condition requires that the influence of $\boldsymbol{\tau}_f$ vanishes for all inputs not in $\mathcal{D}_f$. However, the term $\boldsymbol{\tau}_f^\top \nabla_{\boldsymbol{\theta}} f(\boldsymbol{x}; \boldsymbol{\theta}_o)$ is generally non-zero for arbitrary $\boldsymbol{x} \in \mathcal{D}_r$, since neither $\boldsymbol{\tau}_f$ nor the gradient are guaranteed to be localized. That is, the linearized update will affect predictions on $\mathcal{D}_r$, unless $\nabla_{\boldsymbol{\theta}} f(\boldsymbol{x}; \boldsymbol{\theta}_o)$ itself vanishes for $\boldsymbol{x} \in \mathcal{D}_r$, or unless $\boldsymbol{\tau}_f$ lies in the nullspace of these gradients.

Therefore, we conclude that both the standard task vector approach (Ilharco et al., 2022) and the linearized task vector method (Ortiz-Jimenez et al., 2024) fail to ensure weight disentanglement for ideal unlearning.

# B. Formal Version and Proof of Theorem 3.1

Recent studies (Georgiev et al., 2024) suggest that the optimal stopping epoch in Machine Unlearning (MU) is not universally consistent across different samples within a forgetting set. Building upon this observation, we formalize the insufficiency of a single-model approach due to the stochastic misalignment of optimal unlearning states.

**Theorem B.1** (Formal Version of Theorem 3.1)**.** *Let $\mathcal{D}_f$ be the forgetting dataset with cardinality $N = |\mathcal{D}_f|$. Suppose the unlearning process spans $T$ discrete epochs. For each model $m$ and each sample $i \in \{1, \ldots, N\}$, let $F_{m,i} \in \{1, \ldots, T\}$ be a random variable representing the epoch where sample $i$ is optimally unlearned. Assume $F_{m,i}$ are i.i.d. and uniformly distributed such that $F_{m,i} \sim \mathcal{U}\{1, \ldots, T\}$. A model $m$ is defined as **successful** if there exists a common epoch $t \in \{1, \ldots, T\}$ such that $F_{m,1} = F_{m,2} = \cdots = F_{m,N} = t$. Then, to ensure a success probability of at least $1 - \delta$ for $\delta \in (0, 1]$, the required number of independent models $k$ must satisfy:*

$$k \geq k(\delta) = \left\lceil \frac{\ln \delta}{\ln(1 - T^{1-N})} \right\rceil. \tag{13}$$

*Proof.* Consider a single unlearning model. Let $E$ be the event that the model is successful. By the assumption of independence across samples, the probability that all $N$ samples reach their optimal unlearning state at a specific epoch $t$ is:

$$\Pr(F_1 = t, \ldots, F_N = t) = \prod_{i=1}^{N} \Pr(F_i = t) = \left(\frac{1}{T}\right)^N. \tag{14}$$

Since the success event $E$ occurs if *any* epoch $t \in \{1, \ldots, T\}$ satisfies this condition, and since the epochs are disjoint events, the total success probability $p$ for one model is:

$$p = \Pr\left(\bigcup_{t=1}^{T} F_1 = \cdots = F_N = t\right) = \sum_{t=1}^{T} \left(\frac{1}{T}\right)^N = T \cdot T^{-N} = T^{1-N}. \tag{15}$$

The probability that a single model fails is thus $1 - p = 1 - T^{1-N}$. For $k$ independent and identically distributed models, the probability that all $k$ models fail to reach a successful state is $(1 - p)^k$. To achieve a confidence level of at least $1 - \delta$, we require the failure probability to be bounded by $\delta$:

$$(1 - p)^k \leq \delta. \tag{16}$$

Taking the natural logarithm on both sides (noting that $\ln(1 - p) < 0$):

$$k \ln(1 - p) \leq \ln \delta \implies k \geq \frac{\ln \delta}{\ln(1 - p)}. \tag{17}$$

Substituting $p = T^{1-N}$ into the inequality:

$$k \geq \frac{\ln \delta}{\ln(1 - T^{1-N})}. \tag{18}$$

Since $k$ must be an integer, we take the ceiling function, yielding the required bound:

$$k(\delta) = \left\lceil \frac{\ln \delta}{\ln(1 - T^{1-N})} \right\rceil. \tag{19}$$

This completes the proof. $\qquad\square$

In practice, optimal stopping epochs across samples are likely correlated. We would like to clarify that the i.i.d. uniform assumption is only used to derive a lower bound (of the models needed) for exposition. Our intention is not to provide a tight bound, but to demonstrate the fundamental insufficiency of relying on a single model. The key mechanism behind our result is not independence or uniformity, but misalignment of optimal stopping epochs across forgetting samples.

The theorem can be stated without any i.i.d. assumption. Let $E_m$ denote the event that model $m$ admits a single epoch jointly optimal for all forgetting samples, and define $p_m = Pr(E_m)$. Then,

$$\Pr\left(\bigcup_{m=1}^{k} E_m\right) \geq 1 - (1 - p_\star)^k, \quad p_\star = \inf_m p_m. \tag{20}$$

One can see that if we only have one model, $p_\star$ will be close to zero when the number of forgetting data is large, and it is almost impossible to align these data points. To achieve a success probability of at least $1 - \delta$, it suffices that

$$k \geq \left\lceil \frac{\ln \delta}{\ln(1 - p_\star)} \right\rceil. \tag{21}$$

The original result follows as a special case with $p_\star = T^{1-N}$.

## C. Implementation Details

**Forgetting Scenario.** We focus on random data forgetting and class-wise forgetting in our work. **Random data forgetting** refers to removing a randomly selected subset of training samples, simulating user-level data deletion. In contrast, **class-wise forgetting** removes all samples from specific classes, representing the requirement to erase an entire category of information.

**Baselines.** **RT** retrains the model from scratch using only the retaining dataset $\mathcal{D}_r$. **FT** (Warnecke et al., 2021) fine-tunes the pre-trained model $\theta_o$ on the remaining dataset $\mathcal{D}_r$. **RL** (Graves et al., 2021) fine-tunes the model on the forgetting dataset $\mathcal{D}_f$ using randomly assigned labels to enforce forgetting. **GA** (Thudi et al., 2022) performs gradient ascent on the forgetting data $\mathcal{D}_f$, which often harms the model's utility. **NegGrad+** (Kurmanji et al., 2024a) addresses GA's issue by combining fine-tuning on $\mathcal{D}_r$ and gradient ascent on $\mathcal{D}_f$. **SFRon** (Huang et al., 2025) incorporates the unlearning update into the parameter manifold defined by the retained data, leveraging Hessian modulation that is efficiently approximated through a fast–slow update strategy. **SalUn** (Fan et al., 2023) performs unlearning by optimizing only the salient parameters of the model identified from the random labeled forgetting data. **NegTV** (Ilharco et al., 2022) obtain an unlearning model by linearly subtracting the parameters of the task vector corresponding to the forgetting data from the original model.

**CIFAR-10 on PreResNet-100.** We train the original model and RT model for 200 epochs using the SGD optimizer with a cosine-scheduled learning rate initialized at $0.01$. For the FT, RL, and SalUn methods, they are performed for 10 epochs with a learning rate of $0.01$. The GA and NegGrad+ methods are trained for 5 epochs with a learning rate of $0.01$. The SFRon method is trained for 10 epochs with a learning rate of $0.01$, and forget frequency and alpha is set to 3 and 80 respectively. In the case of NegTV, the model undergoes a finetune model on $\mathcal{D}_f$ for 10 epochs, and the scaling coefficient $\alpha$ is set to 0.9 for random data forgetting and 0.2 for class-wise forgetting. For both $\text{MCU}_\beta$ and MCU, random data forgetting is performed for 10 epochs, whereas class-wise forgetting is conducted for 5 epochs, both with a learning rate of $0.01$.

**ImageNet-100 on ViT.** We utilize a pretrained ViT and fine-tune 30 epochs with a learning rate of $0.001$ to get the original model. The RT method follows the same setting as the original model. For FT, RL, and SalUn, training is performed for 5 epochs, while GA and NegGrad+ are trained for 2 epochs. The SFRon method is trained for 5 epochs with a learning rate of $0.001$, and forget frequency and alpha is set to 3 and 80 respectively. Similarly, the finetuning model for the NegTV method is trained 5 epochs with a learning rate of $0.001$ and a coefficient $\alpha$ of 0.9. For MCUs, they are trained for 2 epochs.

**Tiny-ImageNet on VGG-16-BN.** We train both the original model and the RT model for 100 epochs with a learning rate of $0.1$. The FT, RL, and SalUn methods undergo training for 10 epochs with a learning rate of $0.01$, while the GA and NegGrad+ methods are trained for 5 epochs. The NegTV method finetunes the model on forgetting data $\mathcal{D}_f$ for 10 epochs with a learning rate of $0.01$. The SFRon method is trained for 10 epochs with a learning rate of $0.01$, and forget frequency and alpha is set to 3 and 80 respectively. We observe that increasing the coefficient $\alpha$ of NegTV causes a substantial degradation in both RA and TA. To preserve model performance, we set $\alpha$ to 0.1. For our MCUs (both MCU and $\text{MCU}_\beta$), training is conducted over 5 epochs with a learning rate of $0.01$.

**CIFAR-10 on VGG-16-BN.** Both the original model and the retrained (RT) model were trained for 100 epochs using the SGD optimizer with a cosine-scheduled learning rate initialized at $0.01$. The FT, RL, and SalUn methods were each trained for 10 epochs with a learning rate of $0.01$, while the GA and NegGrad+ methods were trained for 5 epochs using the same learning rate. The SFRon method is trained for 10 epochs with a learning rate of $0.01$, and forget frequency and alpha is set

to 3 and 80 respectively. For NegTV, the model was fine-tuned on $\mathcal{D}_f$ for 10 epochs, with the scaling coefficient $\alpha$ set to 0.9. Both MCU$_\beta$ and MCU were trained for 10 epochs using a learning rate of 0.01.

**Additional MCU Implementation Details.** All our experiments are conducted on a single Tesla V100 GPU. We only use 50% of the retaining data during our MCU training process. The hyperparameters $k$ and $k_r$ are set to 0.5 and 0.1, respectively. For searching the optimal model on the curve, we obtain single models at $t = 0.75$ and 1 first. Then we interpolate to find the optimal model according to the approach in section 3. For searching an effective region, we obtained 20 single models along the pathway.

**MIA Implementation Details.** In line with previous studies (Song et al., 2019; Yeom et al., 2018), we assess the privacy risks of unlearning models using a confidence-driven membership inference attack. We first train a support vector classifier on a balanced dataset, where samples from the retaining data $\mathcal{D}_r$ are labeled as members and those from the test data $\mathcal{D}_{test}$ are labeled as non-members. After training, the attack model is deployed to probe the unlearning model $\theta_u$. To evaluate the unlearning performance, MIA-efficacy is obtained by applying the trained MIA predictor to the unlearning model on the forgetting data. Specifically, MIA-efficacy quantifies the proportion of forgetting data $\mathcal{D}_f$ that the attack correctly rejects as non-members. Formally, MIA-Efficacy $= \frac{TN}{|\mathcal{D}_f|}$, where TN is the number of forgotten samples classified as non-members and $|\mathcal{D}_f|$ is the size of the forgetting data. Under this definition, a higher MIA-efficacy score reflects stronger privacy protection and more complete removal of membership traces from $\theta_u$.

## D. Additional Experimental Results

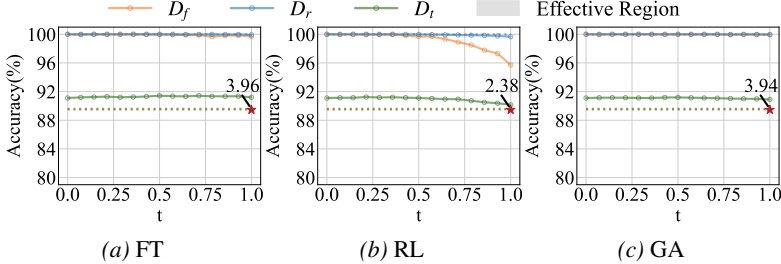

*Figure 9.* Performance of linear interpolation between $\theta_o$ and $\theta_p$ where the pre-unlearning models are FT, RL, and GA, respectively. The results show that linear interpolation fails to produce any effective intermediate unlearning models along the linear pathway. The best-performing model is always the pre-unlearning model itself (i.e., at $t = 1$).

**Linear Interpolation.** The linear interpolation pathway between $\theta_o$ and $\theta_p$ is defined as follows,

$$\phi(t) = (1 - t)\theta_o + t\theta_p, \tag{22}$$

where $t$ represents a scalar interpolation coefficient.

The linear interpolation results on CIFAR-10 with PreResNet-110 under the 10% random data forgetting scenario are shown in Figure 9. As observed, linear interpolation fails to produce any effective intermediate unlearning models along the interpolation pathway. The best-performing model is consistently the pre-unlearning model itself (i.e., at $t = 1$). This behavior can be attributed to the fact that linear interpolation, unlike the Bézier or Polychain curve, does not involve any optimization and therefore cannot bypass loss barriers between $\theta_o$ and $\theta_p$. Existing work (Garipov et al., 2018; Frankle et al., 2020) indicates that linear mode connectivity only holds under specific conditions (e.g., similar learning rates, training trajectories), and does not generally apply across arbitrary model pairs or diverse tasks. Thus, purely linear pathways are not sufficient for discovering intermediate models that balance forgetting quality and retaining performance.

**Additional Unlearning Performance for Baselines and MCUs.** Table 5 shows the results in VGG-16-BN with Tiny-ImageNet under 10% random data forgetting scenario. Table 6 presents the results under 20% random data forgetting scenario across 3 different datasets. We also show additional experimental results conducted in CIFAR-10 with PreResNet-110 as shown in Table 7 and Tiny-ImageNet with VGG-16-BN as shown in Table 8 under the class-wise scenario. Furthermore, the results of 10% random data forgetting on CIFAR-10 with VGG-16-BN are presented in Table 9. These findings consistently align with our previous analysis, further substantiating the effectiveness of our MCUs.

*Table 5.* Unlearning performance of MU methods for **10% random data forgetting** in **Tiny-ImageNet** with **VGG-16-BN**.

| Methods | UA ↓ | RA ↓ | TA ↓ | MIA ↓ | Avg. Gap ↓ | RTE ↓ |
|---|---|---|---|---|---|---|
| RT | $45.45_{\pm0.02}(0.00)$ | $99.52_{\pm0.02}(0.00)$ | $55.59_{\pm0.17}(0.00)$ | $55.79_{\pm0.17}(0.00)$ | 0.00 | 37.46 |
| FT | $5.76_{\pm0.07}(39.69)$ | $99.34_{\pm0.02}(0.18)$ | $56.25_{\pm0.10}(0.66)$ | $15.95_{\pm0.41}(39.84)$ | 20.09 | 3.80 |
| RL | $38.59_{\pm0.25}(6.86)$ | $99.03_{\pm0.02}(0.49)$ | $53.87_{\pm0.32}(1.72)$ | $86.53_{\pm0.29}(30.74)$ | 9.95 | 13.33 |
| GA | $5.17_{\pm0.07}(40.28)$ | $96.11_{\pm0.04}(3.41)$ | $53.66_{\pm0.02}(1.93)$ | $7.89_{\pm0.30}(47.90)$ | 23.38 | 0.32 |
| NegGrad+ | $51.06_{\pm12.91}(5.61)$ | $83.22_{\pm5.81}(16.30)$ | $46.74_{\pm3.00}(8.85)$ | $51.97_{\pm1.30}(3.82)$ | 8.65 | 6.58 |
| SFRon | $28.87_{\pm1.78}(16.58)$ | $97.14_{\pm0.90}(2.38)$ | $53.10_{\pm1.04}(2.49)$ | $15.29_{\pm2.85}(40.50)$ | 15.49 | 12.60 |
| SalUn | $36.61_{\pm0.23}(8.84)$ | $99.03_{\pm0.03}(0.49)$ | $54.04_{\pm0.35}(1.55)$ | $85.37_{\pm0.41}(29.58)$ | 10.12 | 13.79 |
| NegTV | $0.81_{\pm0.01}(44.64)$ | $99.35_{\pm0.02}(0.17)$ | $56.85_{\pm0.03}(1.26)$ | $4.49_{\pm0.20}(51.30)$ | 24.34 | 0.58 |
| MCU | $42.42_{\pm1.23}(3.03)$ | $93.32_{\pm0.33}(6.20)$ | $52.53_{\pm0.15}(3.06)$ | $44.43_{\pm1.41}(11.36)$ | 5.91 | 10.77 |
| MCU-$\beta$ | $49.92_{\pm0.72}(4.47)$ | $92.88_{\pm0.19}(6.64)$ | $52.92_{\pm0.21}(2.67)$ | $46.90_{\pm0.07}(8.89)$ | **5.67** | 10.83 |

Under comprehensive evaluation metrics, both MCU$_\beta$ and MCU consistently rank as the top two performers, achieving results nearly equivalent to the RT model. Notably, MCU$_\beta$ achieves 100% unlearning accuracy on forgetting data, ensuring robust and reliable performance across diverse settings. The results in Tables 5 - 9 further emphasize the limitation of the linear approach, NegTV. Our experimental results of NegTV reveal a significant performance instability for NegTV across different datasets in class-wise forgetting scenarios. While NegTV demonstrates substantial advantages on ImageNet-100 in Table 2, its performance deteriorates considerably on both CIFAR-10 and Tiny-ImageNet datasets, underscoring its lack of robustness. Furthermore, we attempted to optimize NegTV by extensively tuning its scaling coefficient $\alpha$ in our experiments, but encountered a persistent dilemma: the method either resulted in under-forgetting (failing to adequately remove the influence of the forgetting class) or over-forgetting (excessively degrading model performance) in the class-wise forgetting scenario. This extreme phenomenon suggests a weight entanglement issue in the linear NegTV method to achieve the delicate balance required for effective class-wise unlearning. Comparing our MCUs with NegTV, we observe that the nonlinear pathway leads to more stable and effective unlearning. This indicates that our nonlinear unlearning method, MCU, is free from the weight entanglement issue that exists in the linear approach.

**Stability to Scarce Retaining Data.** The amount of retaining data $\mathcal{D}_r$ used during our training process can be only a subset of the full set. The intuition is that the end models $\boldsymbol{\theta}_o$ and $\boldsymbol{\theta}_p$ already preserve sufficient information about $\mathcal{D}_r$. As a result, our framework is able to consistently identify an effective unlearning pathway, making it notably insensitive to scarce retaining data. We validate this claim on CIFAR-10 with PreResNet-100 under the 10% random data forgetting scenario, with NegGrad+ as pre-unlearning model in our MCU framework. As illustrated in Figure 11, the accuracy values of the optimal unlearning model on the MCU pathway remain stable across varying retaining data proportions.

**Superiority of the Adaptive $\beta$ Strategy under Scarce Retaining Data** In Figure 10, we present the results of nonlinear pathway searching across varying proportions of retaining data $\mathcal{D}_r$, ranging from 10% to 100%. These experiments were conducted using MCU$_\beta$ on CIFAR-10 with PreResNet-110 under the 10% random data forgetting scenario. MCU$_\beta$ consistently outperforms other unlearning methods across all retaining data proportion settings (see Table 1 for other baselines' specific results). As expected, the optimal performance is achieved when utilizing 100% of the retaining data for curve training. In this case, the pathway searching process fully leverages the entire dataset, leading to the highest retaining accuracy and minimizing any degradation in model utility. By comparison, the worst performance occurs when only 10% or 20% of the retaining data is available. In these cases, the retaining accuracy drops significantly, indicating that an insufficient amount of retaining data negatively impacts the learning process. However, when the proportion of $\mathcal{D}_r$ exceeds 30%, retaining accuracy remains consistently high with relatively small average accuracy gaps. This demonstrates the inherent stability of our

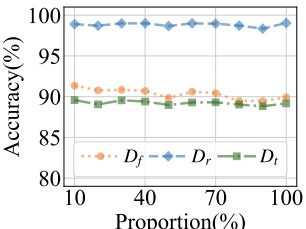

*Figure 11.* The accuracy on $\mathcal{D}_f$, $\mathcal{D}_r$, and $\mathcal{D}_t$ across different proportions of retaining data used in our training process. It shows that all accuracy performance remains stable even with 10% retaining data.

MCUs even under limited retaining data conditions. This stems from our framework of searching nonlinear pathways in the parameter space between the original and pre-unlearning models as end points, which effectively preserves critical retaining data information along the pathway. Consequently, an effective unlearning model can consistently be identified across the pathway, regardless of the scarce retaining data used. Overall, we suggest that maintaining at least 30% of the retaining data

*Table 6.* Overall performance of MU methods for **20% random data forgetting**. The results are presented in the format $a \pm b$, with $a$ as the mean and $b$ as the standard deviation from 5 independent trials. The performance gap relative to RT method is represented in (•). The Avg. Gap is derived by averaging performance gaps across accuracy-related metrics, including UA, RA, TA and MIA. Smaller gaps reflect closer alignment with the RT model's performance. RTE is reported in minutes.

| Methods | UA ↓ | RA ↓ | TA ↓ | MIA ↓ | Avg. Gap ↓ | RTE ↓ |
|---|---|---|---|---|---|---|
| **CIFAR-10 with PreResNet-110** | | | | | | |
| RT | $11.17_{\pm0.08}(0.00)$ | $99.97_{\pm0.01}(0.00)$ | $88.92_{\pm0.17}(0.00)$ | $19.03_{\pm0.28}(0.00)$ | 0.00 | 93.75 |
| FT | $0.34_{\pm0.05}(10.83)$ | $99.94_{\pm0.01}(0.03)$ | $90.89_{\pm0.14}(1.97)$ | $3.43_{\pm0.10}(15.60)$ | 7.11 | 4.72 |
| RL | $3.24_{\pm0.14}(7.93)$ | $99.34_{\pm0.04}(0.63)$ | $90.24_{\pm0.17}(1.32)$ | $23.64_{\pm0.27}(4.61)$ | 3.62 | 7.83 |
| GA | $0.03_{\pm0.00}(11.14)$ | $99.98_{\pm0.00}(0.01)$ | $90.86_{\pm0.01}(1.94)$ | $0.80_{\pm0.05}(18.23)$ | 7.83 | 0.65 |
| NegGrad+ | $5.22_{\pm0.16}(5.95)$ | $98.51_{\pm0.08}(1.46)$ | $89.32_{\pm0.13}(0.40)$ | $10.03_{\pm0.32}(9.00)$ | 4.20 | 2.96 |
| SFRon | $14.70_{\pm0.43}(3.53)$ | $89.08_{\pm0.63}(10.89)$ | $85.65_{\pm0.41}(3.27)$ | $69.28_{\pm1.89}(50.25)$ | 16.99 | 13.87 |
| SalUn | $3.87_{\pm0.23}(7.30)$ | $98.76_{\pm0.04}(1.21)$ | $89.95_{\pm0.13}(1.03)$ | $24.94_{\pm0.49}(5.91)$ | 3.86 | 7.96 |
| NegTV | $3.33_{\pm0.35}(7.84)$ | $98.27_{\pm0.12}(1.70)$ | $86.86_{\pm0.32}(2.06)$ | $6.83_{\pm0.21}(12.20)$ | 5.95 | 1.28 |
| MCU | $7.21_{\pm0.03}(3.96)$ | $98.20_{\pm0.10}(1.77)$ | $88.23_{\pm0.12}(0.69)$ | $13.64_{\pm0.57}(5.39)$ | 2.95 | 6.88 |
| $MCU_\beta$ | $8.04_{\pm0.03}(2.50)$ | $97.90_{\pm0.01}(2.08)$ | $88.68_{\pm0.14}(0.91)$ | $13.42_{\pm0.78}(5.61)$ | **2.78** | 6.92 |
| **ImageNet-100 with ViT** | | | | | | |
| RT | $11.89_{\pm0.00}(0.00)$ | $92.08_{\pm0.00}(0.00)$ | $88.04_{\pm0.04}(0.00)$ | $14.47_{\pm0.05}(0.00)$ | 0.00 | 837.96 |
| FT | $8.87_{\pm0.14}(3.02)$ | $92.32_{\pm0.01}(0.24)$ | $87.75_{\pm0.11}(0.29)$ | $10.53_{\pm0.91}(3.94)$ | 1.87 | 84.88 |
| RL | $9.54_{\pm0.10}(2.35)$ | $91.85_{\pm0.04}(0.23)$ | $87.83_{\pm0.11}(0.21)$ | $29.43_{\pm2.67}(14.96)$ | 4.44 | 245.60 |
| GA | $12.42_{\pm2.08}(0.53)$ | $87.68_{\pm2.25}(4.40)$ | $84.99_{\pm1.81}(3.05)$ | $12.12_{\pm1.05}(2.35)$ | 2.58 | 59.57 |
| NegGrad+ | $12.12_{\pm0.89}(0.23)$ | $91.67_{\pm0.27}(0.41)$ | $86.86_{\pm0.42}(1.18)$ | $15.63_{\pm0.45}(1.16)$ | 0.74 | 85.96 |
| SFRon | $25.77_{\pm0.00}(13.88)$ | $78.50_{\pm0.00}(13.58)$ | $77.86_{\pm0.00}(10.18)$ | $59.40_{\pm1.68}(44.93)$ | 20.64 | 87.65 |
| SalUn | $8.85_{\pm0.88}(3.04)$ | $91.67_{\pm0.29}(0.41)$ | $87.75_{\pm0.29}(0.29)$ | $22.65_{\pm0.00}(8.18)$ | 2.98 | 225.75 |
| NegTV | $10.06_{\pm0.04}(1.83)$ | $91.47_{\pm0.05}(0.61)$ | $87.11_{\pm0.17}(0.93)$ | $13.07_{\pm0.29}(1.40)$ | 1.19 | 22.29 |
| MCU | $11.78_{\pm0.12}(0.11)$ | $91.06_{\pm0.03}(1.02)$ | $87.22_{\pm0.11}(0.82)$ | $14.89_{\pm0.22}(0.42)$ | 0.59 | 150.57 |
| $MCU_\beta$ | $10.98_{\pm0.07}(0.91)$ | $92.06_{\pm0.10}(0.02)$ | $87.40_{\pm0.14}(0.64)$ | $14.58_{\pm0.18}(0.11)$ | **0.42** | 149.88 |
| **Tiny-ImageNet with VGG-16-BN** | | | | | | |
| RT | $46.72_{\pm0.25}(0.00)$ | $99.65_{\pm0.01}(0.00)$ | $54.10_{\pm0.06}(0.00)$ | $57.81_{\pm0.03}(0.00)$ | 0.00 | 33.73 |
| FT | $5.44_{\pm0.03}(41.28)$ | $99.44_{\pm0.01}(0.21)$ | $56.53_{\pm0.11}(2.43)$ | $15.85_{\pm0.16}(41.96)$ | 21.47 | 4.20 |
| RL | $30.49_{\pm0.39}(16.23)$ | $98.77_{\pm0.03}(0.88)$ | $52.61_{\pm0.19}(1.49)$ | $83.52_{\pm0.45}(25.71)$ | 11.08 | 14.11 |
| GA | $4.42_{\pm0.14}(42.30)$ | $95.91_{\pm0.13}(3.74)$ | $53.60_{\pm0.08}(0.50)$ | $7.83_{\pm0.19}(49.98)$ | 24.13 | 0.50 |
| NegGrad+ | $45.02_{\pm0.70}(1.71)$ | $85.23_{\pm0.31}(14.42)$ | $47.55_{\pm0.27}(6.55)$ | $40.13_{\pm0.11}(17.68)$ | 10.09 | 4.22 |
| SFRon | $33.12_{\pm1.22}(13.60)$ | $97.94_{\pm0.41}(1.71)$ | $52.27_{\pm1.40}(1.83)$ | $23.20_{\pm3.15}(34.61)$ | 12.94 | 20.56 |
| SalUn | $39.55_{\pm0.01}(7.17)$ | $97.66_{\pm0.04}(1.99)$ | $53.32_{\pm0.29}(0.78)$ | $86.07_{\pm0.36}(28.26)$ | 9.55 | 13.73 |
| NegTV | $1.85_{\pm0.96}(44.87)$ | $98.81_{\pm0.54}(0.84)$ | $56.04_{\pm0.69}(1.94)$ | $6.95_{\pm1.83}(50.86)$ | 24.63 | 0.97 |
| MCU | $38.38_{\pm0.09}(8.34)$ | $97.73_{\pm0.18}(1.92)$ | $52.35_{\pm0.12}(1.75)$ | $47.25_{\pm1.12}(10.56)$ | 5.64 | 9.78 |
| $MCU_\beta$ | $44.72_{\pm0.06}(2.00)$ | $96.94_{\pm0.07}(2.71)$ | $50.75_{\pm0.25}(3.35)$ | $45.25_{\pm0.50}(12.56)$ | **5.16** | 8.44 |

during pathway searching is enough to achieve a balance between training efficiency, effective unlearning, and model utility.

**Robustness of Adaptive $\beta$ Strategy to Hyperparameters $k$ and $k_r$.** While $MCU_\beta$ under the default settings of $k = 0.5$ and $k_r = 0.1$ already yield strong performance in the main paper's experimental results, we further investigate the robustness of our proposed adaptive unlearning penalty coefficient $\beta$ under different values of $k$ and $k_r$. Specifically, we conduct ablation studies over a range of values: $k \in \{0.3, 0.5, 0.8\}$ and $k_r \in \{0.0, 0.1, 0.2\}$.

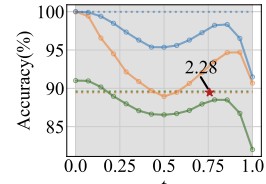

*Figure 12.* $MCU_\beta$ performance with an extremely poor GA pre-unlearning model.

Figures 13 and 14 present the results of these experiments. In each figure, subplots (a)-(c) show results for the vanilla MCU with a fixed penalty coefficient $\beta = 0.2$, while (d)-(f) show the results for $MCU_\beta$ with our adaptive strategy. In Figure 13, $k_r$ is fixed at 0.1 while varying $k$, and in Figure 14, $k$ is fixed at 0.5 while varying $k_r$.

The results demonstrate that $MCU_\beta$ exhibits strong robustness to changes in both $k$ and $k_r$. Its performance remains stable across different settings, indicating that the adaptive penalty strategy effectively accommodates varying $k$ and $k_r$. In contrast, the vanilla MCU model shows noticeable fluctuations, suggesting a greater sensitivity to hyperparameter choices. This highlights the advantage of using an adaptive $\beta$ for more reliable unlearning performance under diverse conditions.

*Table 7.* Unlearning performance of MU methods for **class-wise forgetting** on **CIFAR-10** with **PreResNet-110**. The table adopts the same format as Table 7.

| Methods | UA $\downarrow$ | UA$_{test}$ $\downarrow$ | RA $\downarrow$ | TA $\downarrow$ | MIA $\downarrow$ | Avg. Gap $\downarrow$ | RTE $\downarrow$ |
|---|---|---|---|---|---|---|---|
| **Class-wise Forgetting** | | | | | | | |
| RT | $100.00_{\pm0.00}(0.00)$ | $100.00_{\pm0.00}(0.00)$ | $99.98_{\pm0.00}(0.00)$ | $90.37_{\pm0.08}(0.00)$ | $100.00_{\pm0.00}(0.00)$ | 0.00 | 104.93 |
| FT | $18.53_{\pm1.65}(81.47)$ | $24.63_{\pm2.75}(75.37)$ | $99.94_{\pm0.02}(0.04)$ | $91.01_{\pm0.10}(0.64)$ | $43.18_{\pm3.26}(56.82)$ | 42.87 | 5.52 |
| RL | $100.00_{\pm0.00}(0.00)$ | $100.00_{\pm0.00}(0.00)$ | $96.67_{\pm0.45}(3.31)$ | $87.81_{\pm0.61}(2.56)$ | $100.00_{\pm0.00}(0.00)$ | 1.17 | 6.80 |
| GA | $85.01_{\pm0.19}(14.99)$ | $88.50_{\pm0.08}(11.50)$ | $90.55_{\pm0.40}(9.43)$ | $80.90_{\pm0.40}(9.47)$ | $86.27_{\pm0.07}(13.73)$ | 11.82 | 0.40 |
| NegGrad+ | $99.94_{\pm0.05}(0.06)$ | $100.00_{\pm0.00}(0.00)$ | $98.07_{\pm0.25}(1.91)$ | $87.25_{\pm0.28}(3.12)$ | $99.97_{\pm0.04}(0.03)$ | 1.02 | 2.95 |
| SFRon | $100.00_{\pm0.00}(0.00)$ | $100.00_{\pm0.00}(0.00)$ | $83.46_{\pm0.34}(16.52)$ | $81.80_{\pm0.37}(8.57)$ | $100.00_{\pm0.00}(0.00)$ | 5.02 | 18.59 |
| SalUn | $100.00_{\pm0.00}(0.00)$ | $100.00_{\pm0.00}(0.00)$ | $99.81_{\pm0.01}(0.17)$ | $90.34_{\pm0.30}(0.03)$ | $100.00_{\pm0.00}(0.00)$ | 0.04 | 6.97 |
| NegTV | $25.28_{\pm4.92}(74.72)$ | $31.95_{\pm5.35}(68.05)$ | $93.03_{\pm0.04}(6.95)$ | $82.43_{\pm0.22}(7.94)$ | $29.05_{\pm3.75}(70.95)$ | 45.72 | 0.71 |
| MCU | $99.96_{\pm0.01}(0.04)$ | $100.00_{\pm0.00}(0.00)$ | $99.80_{\pm0.01}(0.18)$ | $90.37_{\pm0.03}(0.00)$ | $100.00_{\pm0.00}(0.00)$ | 0.04 | 7.27 |
| MCU$_\beta$ | $100.00_{\pm0.00}(0.00)$ | $100.00_{\pm0.00}(0.00)$ | $99.85_{\pm0.00}(0.13)$ | $90.37_{\pm0.03}(0.00)$ | $100.00_{\pm0.00}(0.00)$ | **0.03** | 7.29 |

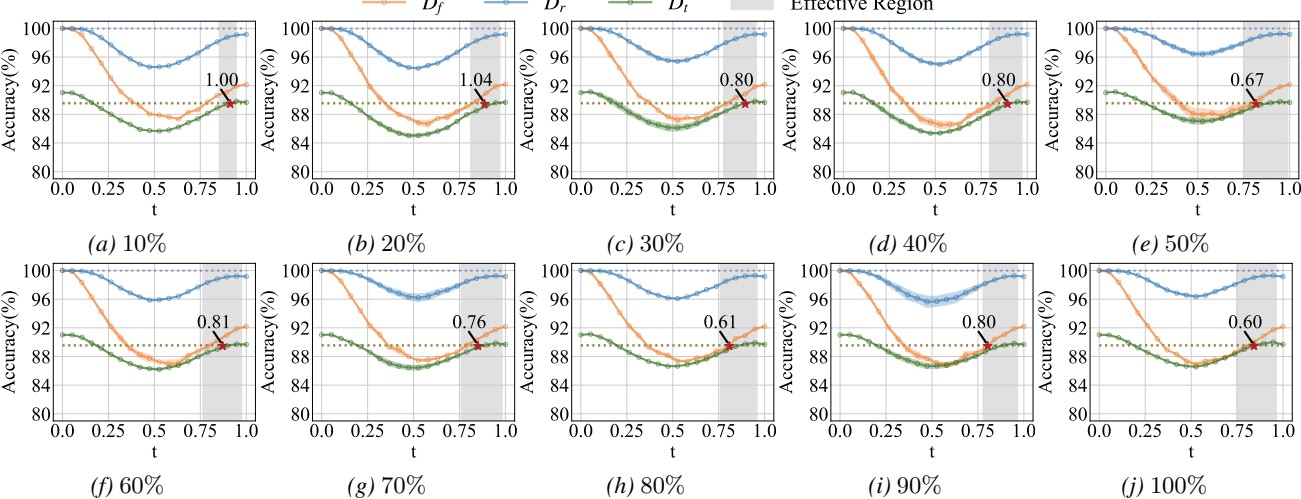

*Figure 10.* Performance with different proportions of retaining data in pathway searching process. The results show that MCU$_\beta$ consistently outperforms other unlearning methods (see Table 1 for the specific values of all baselines) across all retaining data proportion settings.

*Table 8.* Unlearning performance of MU methods for **class-wise forgetting** on **TinyImageNet** with **VGG-16-BN**.

| Methods | UA ↓ | UA$_{test}$ ↓ | RA ↓ | TA ↓ | MIA ↓ | Avg. Gap ↓ | RTE ↓ |
|---|---|---|---|---|---|---|---|
| | | | **Class-wise Forgetting** | | | | |
| RT | $100.00_{\pm 0.00}(0.00)$ | $100.00_{\pm 0.00}(0.00)$ | $99.34_{\pm 0.03}(0.00)$ | $56.94_{\pm 0.11}(0.00)$ | $100.00_{\pm 0.00}(0.00)$ | 0.00 | 42.06 |
| FT | $74.27_{\pm 2.45}(25.73)$ | $78.67_{\pm 5.25}(21.33)$ | $99.29_{\pm 0.02}(0.05)$ | $56.71_{\pm 0.12}(0.23)$ | $90.53_{\pm 2.07}(9.47)$ | 11.36 | 4.29 |
| RL | $98.87_{\pm 1.04}(1.13)$ | $100.00_{\pm 0.00}(0.00)$ | $98.83_{\pm 0.01}(0.51)$ | $56.52_{\pm 0.10}(0.42)$ | $100.00_{\pm 0.00}(0.00)$ | 0.41 | 7.24 |
| GA | $91.80_{\pm 0.59}(8.20)$ | $87.33_{\pm 0.94}(12.67)$ | $94.75_{\pm 0.06}(4.59)$ | $52.86_{\pm 0.05}(4.08)$ | $96.60_{\pm 0.16}(3.40)$ | 6.59 | 0.13 |
| NegGrad+ | $94.76_{\pm 1.50}(5.24)$ | $93.60_{\pm 3.67}(6.40)$ | $99.33_{\pm 0.03}(0.01)$ | $56.73_{\pm 0.06}(0.21)$ | $97.33_{\pm 1.97}(2.67)$ | 2.91 | 2.25 |
| SFRon | $100.00_{\pm 0.00}(0.00)$ | $100.00_{\pm 0.00}(0.00)$ | $88.62_{\pm 0.07}(10.72)$ | $51.90_{\pm 0.21}(5.04)$ | $100.00_{\pm 0.00}(0.00)$ | 3.15 | 12.03 |
| SalUn | $99.27_{\pm 0.62}(0.73)$ | $100.00_{\pm 0.00}(0.00)$ | $98.95_{\pm 0.02}(0.39)$ | $56.58_{\pm 0.15}(0.36)$ | $100.00_{\pm 0.00}(0.00)$ | 0.30 | 7.25 |
| NegTV | $0.50_{\pm 0.10}(99.50)$ | $50.00_{\pm 0.00}(50.00)$ | $99.38_{\pm 0.02}(0.04)$ | $56.96_{\pm 0.00}(0.02)$ | $6.10_{\pm 0.90}(93.9)$ | 48.69 | 0.20 |
| MCU | $100.00_{\pm 0.00}(0.00)$ | $100.00_{\pm 0.00}(0.00)$ | $99.10_{\pm 0.01}(0.24)$ | $56.44_{\pm 0.02}(0.50)$ | $100.00_{\pm 0.00}(0.00)$ | **0.15** | 5.78 |
| MCU$_\beta$ | $100.00_{\pm 0.00}(0.00)$ | $100.00_{\pm 0.00}(0.00)$ | $99.07_{\pm 0.01}(0.27)$ | $56.47_{\pm 0.09}(0.47)$ | $100.00_{\pm 0.00}(0.00)$ | **0.15** | 5.77 |

*Table 9.* Unlearning performance of MU methods for **10% random data forgetting** scenario on **CIFAR-10** with **VGG-16-BN**.

| Methods | UA ↓ | RA ↓ | TA ↓ | MIA ↓ | Avg. Gap ↓ | RTE ↓ |
|---|---|---|---|---|---|---|
| RT | $10.00_{\pm 0.12}(0.00)$ | $99.96_{\pm 0.01}(0.00)$ | $89.81_{\pm 0.07}(0.00)$ | $15.14_{\pm 0.02}(0.00)$ | 0.00 | 17.97 |
| FT | $0.25_{\pm 0.04}(9.75)$ | $99.94_{\pm 0.02}(0.02)$ | $90.31_{\pm 0.08}(0.50)$ | $1.31_{\pm 0.11}(13.83)$ | 6.03 | 1.84 |
| RL | $14.98_{\pm 0.60}(4.98)$ | $99.90_{\pm 0.01}(0.06)$ | $88.53_{\pm 0.05}(1.28)$ | $57.50_{\pm 0.86}(42.36)$ | 12.17 | 7.38 |
| GA | $0.24_{\pm 0.00}(9.76)$ | $99.93_{\pm 0.00}(0.03)$ | $87.67_{\pm 0.02}(2.14)$ | $4.20_{\pm 0.11}(10.94)$ | 5.72 | 0.17 |
| NegGrad+ | $2.63_{\pm 0.25}(7.37)$ | $99.77_{\pm 0.03}(0.19)$ | $89.88_{\pm 0.21}(0.07)$ | $4.57_{\pm 1.84}(10.57)$ | 4.55 | 2.52 |
| SFRon | $20.48_{\pm 1.66}(10.48)$ | $88.46_{\pm 1.80}(11.50)$ | $84.09_{\pm 0.81}(5.72)$ | $26.28_{\pm 1.20}(11.14)$ | 9.71 | 6.23 |
| SalUn | $11.30_{\pm 0.24}(1.30)$ | $99.34_{\pm 0.02}(0.62)$ | $89.66_{\pm 0.20}(0.15)$ | $22.50_{\pm 1.07}(7.36)$ | 2.36 | 5.96 |
| NegTV | $4.58_{\pm 0.06}(5.42)$ | $98.06_{\pm 0.10}(1.90)$ | $85.00_{\pm 0.09}(4.81)$ | $5.23_{\pm 0.37}(9.91)$ | 5.51 | 0.32 |
| MCU | $9.71_{\pm 0.28}(0.29)$ | $99.14_{\pm 0.06}(0.82)$ | $88.22_{\pm 0.03}(1.59)$ | $14.73_{\pm 0.12}(0.41)$ | **0.77** | 5.89 |
| MCU$_\beta$ | $9.99_{\pm 0.01}(0.01)$ | $99.58_{\pm 0.01}(0.38)$ | $88.31_{\pm 0.09}(1.50)$ | $14.55_{\pm 0.07}(0.59)$ | **0.62** | 5.88 |

**1% Random Data Forgetting.** To evaluate the effectiveness of our MCUs in scenarios with extremely sparse forgetting data, we conduct experiments with 1% random data forgetting on CIFAR-10 using PreResNet-110. As shown in Table 10, our MCUs still perform well under this challenging setting.

**Performance of Different Parameter Mask Strategies.** In this section, we analyze alternative parameter mask strategies proposed by other machine unlearning methods, specifically SalUn and SFRon. All experiments are conducted on the CIFAR-10 dataset using PreResNet-110 under the 10% random data forgetting setting. We examine the effectiveness of different parameter mask strategies by substituting the masks used in SalUn (Fan et al., 2023) and SFRon (Huang et al., 2025) into our MCU$_\beta$ framework. The results are shown in Table 11. We observe that MCU$_\beta$ remains effective regardless of the specific masking strategy applied, indicating the robustness of our framework. The RA and TA results of MCU$_\beta - m_{\text{SalUn}}$ are relatively poor because the SalUn mask only considers the importance of parameters to the forgetting data, without accounting for the need to freeze parameters important to the retaining data. This leads to poor performance in both RA and TA. However, our masking strategy offers a significant advantage in terms of running time efficiency (RTE). While element-wise parameter masks (as used in SFRon and SalUn) still require gradient computation for all parameters during training, limiting practical speedup, our method masks entire parameters. This allows us to completely bypass gradient computations for masked parameters during training, resulting in substantial runtime improvements.

**Robustness under Extreme Pre-Unlearning Conditions.** One common concern regarding our MCU framework is its potential dependence on the quality of the pre-unlearning model. To rigorously evaluate the robustness of our approach, we designed an experiment using GA as the pre-unlearning method and intentionally pushed it to an extremely poor performance regime. GA is well-known for its instability, high variance, and tendency to severely disrupt model parameters.

In particular, we trained GA for 10 epochs to deliberately induce gradient explosion, which drastically degraded the RA and severely impaired the model's utility. As shown in Figure 12, even under this highly unfavorable initialization, our MCU$_\beta$ was still able to identify a valid unlearning pathway that not only improves the forgetting performance but also preserves a substantially large effective unlearning region. These findings confirm that MCU$_\beta$ remains highly effective even when

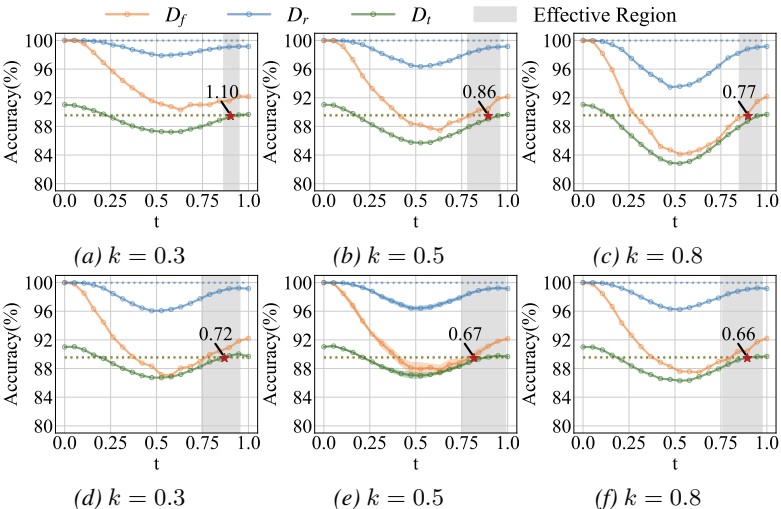

*Figure 13.* Robustness analysis of $k$ on MCU and MCU$_\beta$. (a)-(c) are results of MCU while (d)-(f) are results of MCU$_\beta$. Compared to vanilla MCU, MCU$_\beta$ demonstrates greater robustness to variations in $k$.

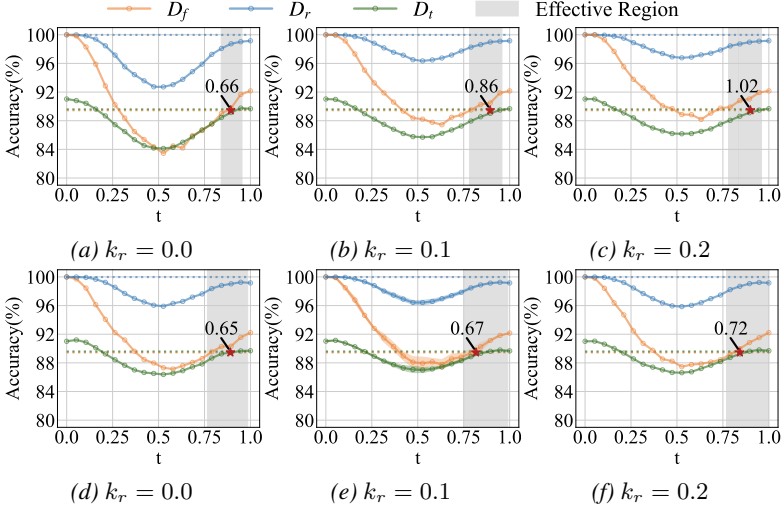

*Figure 14.* Robustness analysis of $k_r$ on MCU and MCU$_\beta$. (a)-(c) are results of MCU while (d)-(f) are results of MCU$_\beta$. Compared to vanilla MCU, MCU$_\beta$ demonstrates greater robustness to variations in $k_r$.

starting from a severely compromised pre-unlearning model, highlighting its robustness under worst-case conditions.

**Comparing with Best Intermediate Checkpoints of Pre-unlearning Models.**
To address concerns about the fairness of our model selection strategy, we compare MCU with baseline methods that also apply post-hoc checkpoint selection. Specifically, we apply a similar best-$t$ search strategy (as used in MCU) to identify the best intermediate checkpoint from the training trajectory of pre-unlearning models. Revisit the cases of under-forgetting (RL with 15 epochs) and over-forgetting (RL with 20 epochs) that we discussed in Figure 4. For each case, we also examine the training checkpoints of pre-unlearning models and apply a similar best-$t$ search strategy used in MCU to identify the checkpoint that minimizes the average gap. The experiments are conducted on CIFAR-10 with PreResNet-110 under 10% random data forgetting scenario.

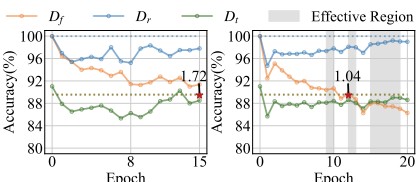

*(a)* Under-forgetting  *(b)* Over-forgetting

*Figure 15.* Best intermediate checkpoints on under-forgetting and over-forgetting pre-unlearning model $\theta_p$.

The accuracy values at epoch=0 are the original model's results and values at epoch=0 are the original model's results. For the under-forgetting scenario, the result is shown in Figure 15a. Despite applying our best-$t$ search strategy to the training trajectory, we are consistently unable to find any intermediate checkpoint that achieves a better average gap than the final

*Table 10.* Unlearning performance of MU methods for **1% random data forgetting** scenario on **CIFAR-10** with **PreResNet-110**.

| Methods | UA ↓ | RA ↓ | TA ↓ | MIA ↓ | Avg. Gap ↓ | RTE ↓ |
|---|---|---|---|---|---|---|
| RT | $11.00_{\pm0.01}(0.00)$ | $99.98_{\pm0.00}(0.00)$ | $90.66_{\pm0.02}(0.00)$ | $18.20_{\pm0.07}(0.00)$ | 0.00 | 78.80 |
| FT | $0.47_{\pm0.34}(10.53)$ | $99.93_{\pm0.01}(0.05)$ | $90.96_{\pm0.15}(0.30)$ | $3.20_{\pm0.58}(15.00)$ | 6.47 | 3.99 |
| RL | $17.20_{\pm1.56}(6.20)$ | $99.91_{\pm0.01}(0.07)$ | $90.45_{\pm0.05}(0.21)$ | $46.6_{\pm0.65}(28.40)$ | 8.72 | 8.98 |
| GA | $1.40_{\pm1.56}(9.60)$ | $99.18_{\pm1.09}(0.80)$ | $90.12_{\pm1.04}(0.54)$ | $3.53_{\pm2.11}(14.67)$ | 6.40 | 0.08 |
| NegGrad+ | $20.47_{\pm0.98}(9.47)$ | $98.37_{\pm0.07}(1.61)$ | $89.68_{\pm0.32}(0.98)$ | $24.80_{\pm1.77}(6.60)$ | 4.67 | 3.06 |
| SFRon | $39.20_{\pm7.20}(28.20)$ | $62.41_{\pm8.40}(37.57)$ | $62.88_{\pm8.36}(27.78)$ | $32.10_{\pm2.24}(13.90)$ | 26.86 | 16.11 |
| SalUn | $7.53_{\pm0.50}(3.47)$ | $99.05_{\pm0.03}(0.93)$ | $91.03_{\pm0.03}(0.37)$ | $17.47_{\pm1.05}(0.73)$ | 1.38 | 6.17 |
| NegTV | $0.13_{\pm0.19}(10.87)$ | $99.98_{\pm0.00}(0.00)$ | $90.97_{\pm0.09}(0.31)$ | $1.26_{\pm0.34}(16.94)$ | 7.03 | 0.17 |
| MCU | $10.63_{\pm0.03}(0.37)$ | $99.98_{\pm0.02}(0.00)$ | $89.82_{\pm0.05}(0.84)$ | $17.24_{0.51\pm}(0.96)$ | **0.54** | 6.43 |
| MCU$_\beta$ | $11.02_{\pm0.01}(0.02)$ | $99.97_{\pm0.01}(0.01)$ | $91.02_{\pm0.04}(0.36)$ | $19.00_{0.34\pm}(0.80)$ | **0.30** | 6.43 |

*Table 11.* Different parameter mask strategies for unlearning performance of MCU$_\beta$ on **CIFAR-10** with **PreResNet-110** under **10% random data forgetting**.

| Methods | UA ↓ | RA ↓ | TA ↓ | MIA ↓ | Avg. Gap ↓ | RTE ↓ |
|---|---|---|---|---|---|---|
| RT | $10.54_{\pm0.34}(0.00)$ | $99.98_{\pm0.01}(0.00)$ | $89.59_{\pm0.22}(0.00)$ | $18.41_{\pm0.52}(0.00)$ | 0.00 | 105.70 |
| MCU$_\beta - m_{\text{SalUn}}$ | $10.34_{\pm0.05}(0.20)$ | $98.18_{\pm0.21}(1.80)$ | $88.53_{\pm0.11}(1.06)$ | $16.10_{\pm0.93}(2.31)$ | 1.34 | 9.27 |
| MCU$_\beta - m_{\text{SFRon}}$ | $9.33_{\pm0.06}(1.21)$ | $99.39_{\pm0.06}(0.59)$ | $88.82_{\pm0.19}(0.77)$ | $16.73_{\pm0.43}(1.68)$ | 1.06 | 9.46 |
| MCU$_\beta - m_{\text{Ours}}$ | $10.29_{\pm0.24}(0.25)$ | $98.69_{\pm0.04}(1.29)$ | $89.11_{\pm0.13}(0.48)$ | $16.45_{\pm0.89}(1.96)$ | 1.00 | **6.82** |

checkpoint. This suggests that checkpoint selection cannot compensate for the under-unlearning behavior of the method. Notably, our method achieves an average gap of 0.62, as shown in Figure 4a.

For the over-forgetting scenario, the result is presented in Figure 15b. While we observe that earlier checkpoints can mitigate some of the over-forgetting, even the best checkpoint we identified using the search strategy performs significantly worse than our MCU interpolation (as shown in Figure 4b), i.e., 1.04 vs 0.43. The performance gap remains substantial.

These findings demonstrate that checkpoint selection is **inherently limited by the discrete nature of training snapshots**. Even with optimal post-hoc selection, discrete checkpoints cannot match the performance of MCU's continuous parameter space exploration. This confirms that **MCU's advantage stems from nonlinear interpolation in continuous space, not merely from the selection strategy**. MCU provides a more powerful and flexible mechanism for balancing forgetting and retention than selecting from available training checkpoints.

## E. Practical Cases of Unlearning Pathway

In this section, we present two practical cases to illustrate the significance of our unlearning pathway, which can produce a spectrum of unlearning models.

Consider a scenario involving harmful information removal, where the risk level of a data instance may evolve over time. In real-world applications, certain data may initially be regarded as low-risk, such as hazardous substances, and handled with minimal unlearning to preserve model utility. However, if new findings later reclassify that data as highly sensitive or harmful, a stronger forgetting guarantee becomes necessary to comply with updated safety. In that case, an unlearning model with stronger forgetting quality can be directly obtained from our unlearning pathway.

Similarly, consider a medical image classification scenario (e.g., detecting lung disease from chest X-rays). Certain images may initially be regarded as low-risk because they appear anonymized, and one may choose an unlearning model that prioritizes retaining performance over forgetting quality. However, later findings may reveal that subtle anatomical features, such as bone structure or vascular patterns, can enable re-identification. As a result, these data points may suddenly require a strong forgetting strength to comply with updated privacy standards. In such cases, our approach allows practitioners to directly select a point along the unlearning pathway that offers stronger forgetting quality.

## F. Pseudo Code of MCU Framework

The pseudo code can be found in Algorithm 1. We present it with three components: parameter mask generating, nonlinear pathway searching, and optimal model/effective unlearning region searching.

---

**Algorithm 1** Pseudo code of MCU$_\beta$

---

1: **Hyper-parameters:** number of iterations $n$, learning rate $\eta$, parameter mask parameter $k$ and $k_r$
2: **Require:** original model $\boldsymbol{\theta}_o$, pre-unlearning model $\boldsymbol{\theta}_p$, training accuracy and test accuracy on the original model $\boldsymbol{\theta}_o$
3: *# 1. Generate a parameter mask*
4: Compute loss $\mathcal{L}(\mathcal{D}_r; \boldsymbol{\theta}_o)$ and $\mathcal{L}(\mathcal{D}_f; \boldsymbol{\theta}_o)$
5: Compute gradient $\nabla_{\boldsymbol{\theta}_o}\mathcal{L}(\mathcal{D}_r; \boldsymbol{\theta}_o)$ and $\nabla_{\boldsymbol{\theta}_o}\mathcal{L}(\mathcal{D}_f; \boldsymbol{\theta}_o)$
6: Calculate $\|\nabla_{\boldsymbol{\theta}_o^i}\mathcal{L}(\mathcal{D}_r; \boldsymbol{\theta}_o)\|_2/|\boldsymbol{\theta}_o^i|$ and $\|\nabla_{\boldsymbol{\theta}_o^i}\mathcal{L}(\mathcal{D}_f; \boldsymbol{\theta}_o)\|_2/|\boldsymbol{\theta}_o^i|$ for each parameter
7: Filter out top $k_r$ proportion of parameters based on $\|\nabla_{\boldsymbol{\theta}_o^i}\mathcal{L}(\mathcal{D}_r; \boldsymbol{\theta}_o)\|_2/|\boldsymbol{\theta}_o^i|$ and generate mask $\boldsymbol{m}_r$
8: Preserve top $k$ proportion of parameters based on $\|\nabla_{\boldsymbol{\theta}_o^i}\mathcal{L}(\mathcal{D}_f; \boldsymbol{\theta}_o)\|_2/|\boldsymbol{\theta}_o^i|$ and generate mask $\boldsymbol{m}_f$
9: Calculate parameter mask $\boldsymbol{m} = \mathbb{1}(\boldsymbol{m}_r \,\&\, \boldsymbol{m}_f)$
10: *# 2. Search pathways in parameter space*
11: $\beta \leftarrow 0.5$    (unlearing penalty coefficient is initialized as 0.5)
12: **for** $i \leftarrow 1, 2, ..., n$ **do**
13:     Sample $t \sim U(0, 1)$
14:     Compute accuracy of retaining data and forgetting data
15:     Adaptively update $\beta$ guided by Eq. 7
16:     Compute cross-entropy loss $\mathcal{L}(\mathcal{D}_r; \phi_{\boldsymbol{\theta}_c}(t))$ for retaining data
17:     Compute cross-entropy loss $\mathcal{L}(\mathcal{D}_f; \phi_{\boldsymbol{\theta}_c}(t))$ for forgetting data
18:     Compute MCU loss $\mathcal{L}_{mcu} = \mathcal{L}(\mathcal{D}_r; \phi_{\boldsymbol{\theta}_c}(t)) - \beta \cdot \mathcal{L}(\mathcal{D}_f; \phi_{\boldsymbol{\theta}_c}(t))$
19:     Compute gradient $\nabla_{\boldsymbol{\theta}_c \odot \boldsymbol{m}}\mathcal{L}_{mcu}$ based on the parameter mask $\boldsymbol{m}$
20:     Update $\boldsymbol{\theta}_c$ using gradient descent:
21:        $\boldsymbol{\theta}_c \odot \boldsymbol{m} \leftarrow \boldsymbol{\theta}_c \odot \boldsymbol{m} - \eta\nabla_{\boldsymbol{\theta}_c \odot \boldsymbol{m}}\mathcal{L}_{mcu}$
22: **end for**
23: *# 3. Search optimal model and effective unlearning region on the pathway*
24: Sample $t \sim U(0, 1)$
25: **for** each $t$ **do**
26:     Compute accuracy of retaining data $\mathcal{D}_r$, forgetting data $\mathcal{D}_f$ and test data $\mathcal{D}_t$
27:     Calculate retaining gap, forgetting gap and test gap and their average gap
28:     Compare average gap with pre-unlearning model $\boldsymbol{\theta}_p$ and search the optimal model and effective unlearning models
29: **end for**
30: **Return:** The optimized pathway $\phi_{\boldsymbol{\theta}_c}(t)$ which connects $\boldsymbol{\theta}_o$ and $\boldsymbol{\theta}_p$, optimal unlearning model $\boldsymbol{\theta}_u^*$ and a range of $t$ where can generate effective unlearning models $\boldsymbol{\theta}_u$ across pathway

---

## G. The Usage Statement of Large Language Models

We used Large Language Model (LLMs) only to assist with language polishing. All ideas, methods, and experiments were conceived and implemented by the authors.

