# OpenReview forum: "Exploring Nonlinear Pathway in Parameter Space for Machine Unlearning"
_ICML.cc/2026/Conference — ICML 2026 regular_

### Official Review · Reviewer_ZL8G · 2026-03-10

**Soundness:** 3
**Presentation:** 3
**Significance:** 3
**Originality:** 3
**Overall Recommendation:** 5
**Confidence:** 3

**Summary:**

This paper proposes the Machine Unlearning via Continuous pathways (MCU) framework. The authors suggest using a quadratic Bézier curve to construct a nonlinear pathway in the parameter space between an original model \(\theta_o\) and a "pre-unlearning" model \(\theta_p\) (generated by any existing MU method). They introduce a parameter masking strategy and an adaptive penalty coefficient \(\beta\) to optimize a control model \(\theta_c\) along this pathway, ultimately allowing the user to select an optimal unlearned model from a continuous spectrum.

While the paper introduces a highly creative conceptual framework and addresses real pain points in unlearning, it suffers from compounding computational inefficiencies, heavy reliance on arbitrary heuristics, and a fundamental paradox in its "plug-and-play" design that undermines its practical utility.

**Compliance With Llm Reviewing Policy:**

Affirmed.

**Final Justification:**

My concerns have been addressed; I maintain my score at 5.

**Key Questions For Authors:**

**1. Rigorous Measurement of "Forgetting" vs. "Masking":**
Continuous interpolation raises a critical security question: is the model genuinely unlearning the data, or is the pathway simply suppressing the logits/activations for the forget set (masking)?
*   How does the optimal model on the MCU pathway perform against rigorous Membership Inference Attacks (MIA)?
*   Furthermore, how close is the output distribution (or parameter space) of the optimal MCU model to the "gold standard" of unlearning—a model completely retrained from scratch without the forget set?

**2. The True Cost-Benefit vs. Retraining:**
The MCU framework requires the original model \(\theta_o\), the generation of a pre-unlearning model \(\theta_p\), and the optimization of the control model \(\theta_c\).
*   At what percentage of data removal (e.g., forgetting 10%, 20%, 50% of the dataset) does the cumulative computational cost of the MCU pipeline exceed the cost of simply retraining the model from scratch?
*   A break-even analysis comparing the total FLOPs of MCU versus retraining from scratch would make the practical utility of this method much clearer.

**3. Dependence on the Quality of \(\theta_p\):**
The paper shows that MCU can rescue a degraded \(\theta_p\) (e.g., one ruined by gradient explosion). However, if \(\theta_p\) has suffered from catastrophic forgetting (destroying the retain set accuracy), doesn't the Bézier curve simply shift the selected model closer to \(\theta_o\)? If the optimal point on the curve is forced to stay very close to \(\theta_o\) just to maintain utility, doesn't that inherently mean the model hasn't forgotten enough of the target data? How does the framework ensure that "rescuing" utility isn't just reverting the unlearning process?

**Limitations:**

Yes

**Strengths And Weaknesses:**

**Strengths**
The paper presents an innovative approach to machine unlearning, backed by exceptionally thorough empirical validation in the appendix.
*   **Strong Conceptual Framework:** Applying mode connectivity via Bézier curves to machine unlearning is a clever conceptual leap. Figure 9 in the appendix brilliantly justifies this by proving that simple linear interpolation fails entirely, validating the necessity of the nonlinear pathway.
*   **Rescuing Degraded Models:** One of the most impressive findings is shown in Figure 14. The fact that MCU\(_{\beta}\) can take a pre-unlearning model deliberately ruined by gradient explosion (via GA) and still discover a valid, highly effective unlearning pathway is a strong testament to the method's stability.
*   **Solving the Discrete Checkpoint Limitation:** Figure 15 elegantly proves that MCU's continuous parameter space exploration is fundamentally superior to simply selecting the "best" discrete checkpoint during a standard unlearning training run.
*   **Robustness to Scarce Data:** Figures 10 and 11 demonstrate that MCU remains highly stable even when using only 30% of the retaining data, which helps mitigate some of the computational costs associated with the search phase.

**Weaknesses and Limitations**
While the empirical results are strong, the framework introduces several practical and theoretical limitations that must be acknowledged:
*   **Cumulative Computational Overhead:** MCU is essentially a wrapper method. It requires a pre-unlearning model \(\theta_p\) to be generated *first*, followed by the optimization of the control model \(\theta_c\). Therefore, the total computational cost is strictly greater than the baseline unlearning method alone. While the authors use parameter masking and data subsetting to speed up the \(\theta_c\) search, the pipeline remains inherently more complex and time-consuming than single-shot methods.
*   **Scalability to Massive Architectures:** The experiments are limited to ResNet-110, VGG-16, and ViT on datasets up to ImageNet-100. It remains unclear if finding a smooth Bézier curve between \(\theta_o\) and \(\theta_p\) is feasible or computationally tractable in the parameter space of modern Large Language Models (LLMs) or large diffusion models, which possess billions of parameters.
*   **Coarse-Grained Masking:** To achieve running time efficiency (RTE), the authors note they mask *entire parameters* rather than using element-wise parameter masks (like SalUn and SFRon). While faster, this coarse-grained approach might limit the model's ability to perform highly delicate, fine-grained unlearning tasks without damaging adjacent retained knowledge.

---

> ### Author Rebuttal · Authors · 2026-03-28
>
> # **Response to Reviewer ZL8G**
>
> Thank you for finding our work innovative and for your valuable comments. We address your concerns as follows.
>
> **Supplementary material: https://anonymous.4open.science/r/MCU-ICML26-042D/Supplementary.pdf**
>
> ### **W1: Cumulative Computational Overhead**
>
> **Modest overhead.** Taking Table 1 as an example, MCU (6.80 min) has comparable runtime to the SOTA baseline SalUn (6.38 min), while achieving ~6× better performance (Avg. Gap: 1.18 vs. 5.99). This demonstrates that MCU delivers substantial performance gains without introducing prohibitive computational costs.
>
> **Worthwhile trade-off.** MCU consistently improves all baselines by ~50% in Avg. Gap (Table 3), handling both over- and under-forgetting. This performance gain justifies the additional runtime.
>
> **Unique value.** Unlike baselines that yield a single model, MCU discovers an entire unlearning pathway, enabling practitioners to select models with different forgetting-utility trade-offs as privacy requirements evolve without re-unlearning. This capability makes the additional overhead a worthwhile investment.
>
> ### **W2: Scalability to LLMs**
>
> Due to space limitations, we kindly refer the reviewer to our response to Reviewer whRh for W3 & Q3.
>
> ### **W3: Coarse-Grained Masking**
>
> **Empirical evidence.** Ablation study in Table 11 in Appendix shows MCU remains effective with element-wise masks from SalUn and SFRon, confirming MCU’s robustness to mask granularity.
>
> **Structural justification.** From a structural perspective, if the dominant forgetting directions lie in a low-dimensional subspace (as often observed in over-parameterized models), block-level masking can approximate these directions with limited loss of fidelity. This explains why coarse masking performs competitively in practice.
>
> **Complementary role of the nonlinear pathway.** We emphasize that masking is only one component of our approach. The nonlinear Bézier pathway introduces additional degrees of freedom via the control model, allowing the trajectory to move off the affine subspace and partially compensate for the reduced granularity of masking. Thus, the effective capacity for adjustment is not limited to the mask alone.
>
> ### **Q1: Genuine Unlearning vs. Suppression**
>
> **MIA.** MIA scores are reported in all result tables. MCU achieves competitive MIA scores across all settings, closely aligning with RT baseline (all result tables).
>
> **Output distribution.** Table 15 in our **link** shows MCU consistently reduces KL divergence to the RT baseline on both D_f and D_r compared to θ_p, confirming genuine unlearning rather than suppression.
>
> ### **Q2: Cost-Benefit vs. Retraining**
>
> As the forget ratio increases, RT's training time **decreases** since it only trains on the remaining retaining data D_r. In contrast, MCU's runtime remains relatively **stable** across forget ratios, as it involves both D_r and D_f during pathway searching. So, MCU's runtime will not be influenced by forget ratios aspect too much. Figure 16 in our **link** shows **MCU remains more efficient than retraining across all tested forget ratios up to 50%**, while maintaining low Avg. Gap to RT.
>
> ### **Q3: Dependence on θ_p Quality**
>
> The reviewer's concerns are reasonable, but a distinction needs to be made between two scenarios:
>
> **1. Catastrophic forgetting (extreme case).** If θ_p suffers **complete utility collapse**, recovery becomes fundamentally difficult for any method, not just MCU. Such extreme cases are unrealistic in practice.
>
> **2. Over-forgetting (practical case).** When θ_p suffers from over-forgetting (RA degraded but not collapsed), the optimized Bézier curve does **not** simply revert to θ_o. By optimizing θ_c, the curve navigates a nonlinear trajectory in parameter space that can simultaneously satisfy forgetting and retention objectives — finding regions that a simple linear interpolation cannot reach. As shown in Fig 14, the optimal model is much closer to the θ_p, not θ_o, meaning the model has forgotten effect.
>
> **3. MCU_β⁺.** It is easy to mitigate MCU_β reliance on θ_p by just unfreezing θ_p during unlearning pathway optimization. We denote this variant as MCU_β⁺. Results in Tables 12-13 in the **link** confirm that MCU_β⁺ achieves strong performance even under varying θ_p conditions.
>
> We will further highlight these point in the main text of the revision.

---

> > ### Author Rebuttal · Reviewer_ZL8G · 2026-04-01
> >
> > I believe his reply thoroughly addressed my question. The previous high rating is well-justified, in my opinion.

---

> > > ### Author Response · Authors · 2026-04-01
> > >
> > > We sincerely thank the reviewer for the positive feedback and for the time dedicated to evaluating our manuscript. We are pleased that our responses successfully addressed the concerns.

---

### Official Review · Reviewer_CQsU · 2026-03-11

**Soundness:** 3
**Presentation:** 3
**Significance:** 2
**Originality:** 2
**Overall Recommendation:** 4
**Confidence:** 3

**Summary:**

The authors propose a novel MU framework called Mode Connectivity Unlearning (MCU) that leverages mode connectivity to find an unlearning pathway in a non-linear manner. Additionally, a parameter masking strategy and adaptive adjustment strategy for unlearning penalty coefficient are proposed. The experiments show that the MCU can achieve better performance.

**Compliance With Llm Reviewing Policy:**

Affirmed.

**Final Justification:**

My concerns have been addressed, I would like raise my score to 4.

**Key Questions For Authors:**

1. Can you explain what is the difference between element-level parameter mask and masking an entire parameter, and why the latter improves efficiency?
2. What is the definition of calibration target $Cal$?
3. How do you find the optimal $t$? You should tune the $t$ on a validation set and evaluate on the hold-out set?

**Limitations:**

yes

**Strengths And Weaknesses:**

**Strengths**
1. The proposed Mode Connectivity Unlearning (MCU) framework consistently achieves superior performance across multiple metrics.
2. The use of accuracy-interpolation curve clearly illustrates the effectiveness of the proposed method.
3. The paper provides a rigorous theoretical analysis that a single unlearning model is fundamentally insufficient to cover the misalignment of optimal stopping points for different forgetting data.

**Weaknesses**
1. The novelty is limited. Both pathway searching and parameter masking techniques are well-established concepts in deep learning. This work seems just utilize them in the scenario of machine unlearning. The paper should more clearly articulate the specific innovations made to these techniques to tailor them for unlearning beyond simple implementation.
2. Performance variance on the same random data with 5 independent trials does not sufficiently indicate the effectiveness and stability of unlearning methods. The authors should evaluate the performance on different random data instead.
3. In the provided results (e.g., Table 5 on Tiny-ImageNet with VGG-16), SFRon shows extremely low Unlearning Accuracy (UA: 1.07%) and high average gaps (24.12) compared to other baselines. This performance appears significantly worse than what is typically reported for this baseline. It is crucial to clarify if the hyperparameters for SFRon were extensively tuned for each specific architecture (especially VGG and ViT) or if default settings were used.
4. The writing can be improved (see questions).

---

> ### Author Rebuttal · Authors · 2026-03-28
>
> # **Response to Reviewer CQsU**
>
> We sincerely appreciate your feedback and provide our detailed responses below.
>
> ### **W1: The novelty is limited**
>
> We clarify that our contribution is **not a direct application of existing techniques**, but a **new formulation of machine unlearning as a continuous optimization problem in parameter space**. Prior work treats unlearning as a point estimation problem (i.e., finding a single model), typically via linear parameter updates, which suffer from weight entanglement. In contrast, we reformulate it as learning a nonlinear trajectory, fundamentally changing both the optimization objective and solution space. This enables several non-trivial innovations:
>
> - **Nonlinear Unlearning Paradigm (Conceptual Contribution).** We show linear task arithmetic is fundamentally limited by weight entanglement and propose nonlinear pathways as a principled alternative. Mode connectivity is re-purposed as an optimization mechanism for unlearning, which has not been explored before;
> - **From Single Solution to Solution Manifold.** Existing methods produce a single model. We instead introduce the concept of an **effective unlearning region**, a continuous set of models along the pathway that achieve different forgetting-utility trade-offs. This is more expressive than any point-based method and is theoretically motivated. The necessity of such a spectrum of unlearning models can be found in Appendix E;
> - **Plug-and-Play Improvement.** MCU consistently improves any existing MU method by ~50% in Avg. Gap (Table 3);
> - **Tensor-Level Masking for Pathway Optimization.** Our masking is specifically designed for **pathway learning**, achieving 75% speedup while maintaining effectiveness;
> - **Adaptive β Coupled with Pathway Dynamics.** Our adaptive β is tied to **calibration targets along the pathway**, enabling dynamic balancing across all models on the trajectory, which is unique to our formulation.
>
> Our novelty lies in introducing a **new problem formulation** (pathway-based unlearning) and **designing algorithmic components** tailored to this formulation, rather than reusing existing techniques.
>
> ### **W2: Random seeds**
>
> Our 5 independent trials are exactly conducted by randomly selecting **different 10% subsets** of training data as forget data in each trial, not repeating experiments on the same fixed subset.
>
> ### **W3: SFRon results**
>
> Thank you for pointing this out. SFRon was implemented using the official codebase with default setting. Unlearning performance is highly sensitive to architecture–dataset pairs, and Tiny-ImageNet with VGG-16-BN is particularly challenging due to its relatively low test accuracy.
>
> We further performed a grid search to tune SFRon’s hyperparameters as suggested, with the best results reported in Table 16 (https://anonymous.4open.science/r/MCU-ICML26-042D/Supplementary.pdf).
>
> ### **Q1: Difference between element-level parameter mask and masking an entire parameter.**
>
> Element-level masking selects individual scalars within parameter tensors, but still requires gradients for all parameters during backprop, limiting speedup. In contrast, our tensor-level masking operates on entire tensors (e.g., a convolutional kernel or attention projection matrix). By masking entire tensors, we can completely bypass gradient computation for masked parameters during training, resulting in a substantial 75% runtime reduction as shown in Figure 2.
>
> ### **Q2: Definition of calibration target Cal**
>
> The **calibration target** Cal defines the **reference accuracy** we expect the unlearned model to achieve on forget and retain datasets during training. Specifically:
>
> - For retaining data D_r, Cal(D_r) = Acc_o(D_train), meaning the unlearned model should preserve the original model's training accuracy on retained data;
> - For forgetting data D_f, Cal(D_f) = Acc_o(D_v) in random forgetting, since forgotten data should be treated as if never trained, analogous to unseen validation data, while Cal(D_f) = 0 in class-wise forgetting, since the entire class should be completely forgotten. These calibration targets provide principled reference points for the adaptive β adjustment.
>
> These dataset-specific definitions are formally described in the "**Calibration Principles behind Adaptive β**" in Section 3.4. They provide **principled reference points** that guide the adaptive adjustment of β and the selection of optimal models along the unlearning pathway.
>
> ### **Q3: How do you find the optimal t?**
>
> No validation set tuning is required. The optimal t is found **during inference** at negligible cost. Specifically, we evaluate the accuracies of D_f and D_r on the unlearned model φ_θc(t), then identify t minimizing the average gap between these accuracies and their corresponding **calibration targets** Cal(D_f) and Cal(D_r). As in Section 3.5, we efficiently locate this by evaluating at t=0.75 and t=1 followed by cubic interpolation, avoiding exhaustive search.

---

> > ### Author Rebuttal · Reviewer_CQsU · 2026-04-01
> >
> > Thanks for your rebuttal. My concerns have been addressed, I will raise my score.

---

> > > ### Author Response · Authors · 2026-04-01
> > >
> > > We are very grateful to the reviewer for the positive feedback and for the supportive decision to increase the score. We appreciate the reviewer's insightful comments throughout the review process, which have been invaluable in improving the quality and clarity of our manuscript.

---

### Official Review · Reviewer_7eZU · 2026-03-13

**Soundness:** 3
**Presentation:** 3
**Significance:** 2
**Originality:** 2
**Overall Recommendation:** 4
**Confidence:** 5

**Summary:**

This paper proposes a machine unlearning framework that uses nonlinear mode connectivity to identify a pathway of models that remove the influence of specific training data while mitigating parameter entanglement. The method further improves efficiency and forgetting performance through a parameter masking strategy and an adaptive penalty adjustment, and experiments on image classification show that MCU consistently enhances unlearning effectiveness over existing approaches.

**Compliance With Llm Reviewing Policy:**

Affirmed.

**Key Questions For Authors:**

are there more intuition on why the non linear MU is more advantageous than the other methods that do not follow linear arithmetic operations?


Section 3.4 Adaptive Unlearning ... > Do you think this becomes necessary because the relying on the unlearning model that is obtained from any unlearning method?

Do you think this justification is necessary to ensure that that the $\theta_c$ is not going to perform too bad, which required optimization and parameter setting.  Doesn't this cause additional computation load?

To prove that this isn't additional heavy  computational computation can you provide the time and steps of ablation study that is required for the setting the \beta?

**Limitations:**

Your current impact statement is quite minimal and does not reflect the broader implications of the work. Since the paper proposes a method for removing the influence of private data from trained models, it would be valuable to discuss the potential societal benefits.

**Strengths And Weaknesses:**

Strength:

The paper is very well written, the idea is simply cobnveyed and the paper is very well structured to clearly discuss about the idea and progress of the method to the masking algoirithm.


Weaknesses:

Please compare your method with the SCRUB algorithm (https://arxiv.org/pdf/2302.09880)

line 55, A sample once > Please make an incremental unlearning evaluation of their method against the previous studies.

line 102> right column> the objective of our work> So at the end if we want to operate on the unlearned model which one we would choose? how to aggregate them?

line 138> left column > essential unlearning information ... > what does it mean? essential unlearning information?

line 146> left column > So far from my understanding this pre-unlearning model is dependent on the unlearning method. How you are sure that the $\theta_p$ is a good choice for the unlearning model?

line 151> left column > As a plug-and-play ... > So far from my understanding this pre-unlearning model is dependent on the unlearning method. How you are sure that the $\theta_p$ is a good choice for the unlearning model?


Table 3: From this table,

1. the UA can varry to 6 percent from one algorithm to another algorithm, that is not significant but when comparing with the baselines and you wanted to demonestrate the superiority of your method, even 2 percent was important?


Considering this variation, do you think it is reliable to choose any unlearning method and use it as the $\theta_p$?

To ensure that this experiment is valid, I would like to see an ablation study comparing for example and unlearned model with proper hyperparameters with NegGrad and then another one with another model that has the worst outcome of unlearning by NegGrad and then compare the results MCU on both of them. repeat this experiment for the rest of the methods.

---

> ### Author Rebuttal · Authors · 2026-03-28
>
> # **Response to Reviewer 7eZU**
>
> We sincerely appreciate your feedback, and our detailed responses are provided below.
>
> **Supplementary Material**: https://anonymous.4open.science/r/MCU-ICML26-042D/Supplementary.pdf
>
> ### **W1: Comparison with SCRUB**
>
> Results of SCRUB are reported in Table 14 in the **link**. MCU_β outperforms SCRUB, and further improves SCRUB as a pre-unlearning model.
>
> SCRUB's Rewind mechanism selects the checkpoint closest to a reference forget-set error. We have already reported results under such mechanism in Figure 15 in Appendix, which shows that post-hoc checkpoint selection is **inherently limited by the discrete nature of training snapshots** and fails in under-forgetting scenarios. This confirms that MCU’s advantage stems from nonlinear interpolation in continuous space, not merely from the selection strategy.
>
> ### **W2: line 55, incremental unlearning**
>
> We clarify that this statement does not refer to  incremental unlearning. Instead, it  motivates the need for **a spectrum of unlearning models within a single training run**: as the perceived risk of forgetting data may evolve, practitioners need flexibility to select models with different forgetting strengths without repeated retraining. See Appendix E for further discussion.
>
> ### **W3: line 102>Model selection**
>
> No aggregation is involved. MCU outputs a continuous pathway of candidate models,  but only a single model is selected for deployment. As described in Section 3.5, we selected the optimal model by minimizing the calibration gap relative to target retraining and forgetting performance.  The pathway serves as a search space, not an ensemble.
>
> ### **W4: line 138>"Essential unlearning information"**
>
> What we mean is that θ_p already encodes a **directional shift in parameter space toward forgetting** relative to θ_o. It captures a coarse but informative trajectory of how the model parameters should change to reduce the influence of D_f.
>
> ### **W5 & W6: Quality of θ_p**
>
> We emphasize that θ_p does not need to be optimal. It only needs to provide a forget-oriented direction in parameter space. By “plug-and-play,” we mean that our framework can be seamlessly integrated with existing unlearning methods to enhance their performance. Table 3 shows that MCU_β improves all baselines by ~ 50% in Avg. Gap regardless of their quality, and Figure 14 confirms MCU’s effectiveness even under poor quality. Moreover, the sensitivity of MCU to θ_p can be effectively mitigated by unfreezing θ_p during unlearning pathway optimization. We denote this variant as **MCU_β⁺** (see results in Tables 12-13 in the **link**).
>
> ### **W7: Variation in Table 3**
>
> We respectfully note that UA alone is insufficient. Our primary evaluation metric is Avg. Gap, which jointly measures forgetting (UA and MIA), retraining (RA), and generalization (TA).  Across all methods, MCU_β consistently reduces gaps by ~50% with low variation. As discussed in our response to W5, MCU_β⁺ can further reduce sensitivity to θ_p.
>
> ### **W8: Ablation with different NegGrad settings**
>
> Results with NegGrad under different hyperparameters are in the **link** in Table 12. MCU_β consistently improves all variants by ~50% in Avg. Gap. MCU_β⁺ further reduces sensitivity across all settings.
>
> We note that if θ_p suffers complete model utility collapse (e.g., RA and TA fully degraded), recovery becomes infeasible for any unlearning method. However, such extreme cases are unrealistic in practice, which falls outside the intended use case of any unlearning framework.
>
> ### **Q1: Why nonlinear MU is better**
>
> The key advantage of nonlinear MU is that it transforms unlearning from **a single-model optimization problem into a structured exploration of a continuous solution manifold**. This increased flexibility enables the model to better balance conflicting objectives (forgetting vs. retaining), which cannot be simultaneously satisfied within a single update or trajectory. Consequently, MCU can access low-loss regions of the parameter space that are **unreachable by conventional methods**, leading to improved unlearning performance.
>
> ### **Q2 & Q3: Adaptive β necessity and computation**
>
> **Adaptive β is necessary.** It is introduced primarily for practical usability and convenience, not because MCU relies on a poor pre-unlearning model. Its purpose is to automatically balance forgetting and retraining during optimization, avoiding manual hyperparameter tuning. We emphasize that MCU still achieves good unlearning performance even without adaptive β.
>
> **No extra computation needed.** The adaptive β adjustment relies solely on monitoring training accuracies on D_f and D_r, which are already computed during the forward pass. The runtime overhead of the adaptive β strategy is reflected in all our result tables: the comparison between MCU and MCU_β consistently shows negligible overhead (e.g., 6.80 vs. 6.82 minutes on CIFAR-10).

---

### Official Review · Reviewer_whRh · 2026-03-13

**Soundness:** 3
**Presentation:** 3
**Significance:** 3
**Originality:** 3
**Overall Recommendation:** 4
**Confidence:** 3

**Summary:**

This paper proposes Mode Connectivity Unlearning (MCU), a framework for approximate machine unlearning that explores a nonlinear Bézier curve pathway in parameter space between an original model (θ_o) and a pre-unlearning model (θ_p). The key insight is that linear unlearning methods (task arithmetic) suffer from weight entanglement, while a nonlinear pathway can find intermediate models that better balance forgetting quality and utility retention. The framework includes three components: (1) pathway searching via a quadratic Bézier curve with an optimized control point θ_c, (2) a two-stage parameter mask that filters parameters critical to retaining data and reserves those critical to forgetting data, and (3) an adaptive penalty coefficient β that dynamically adjusts the forgetting-utility tradeoff. Experiments on image classification tasks across multiple datasets and architectures demonstrate consistent improvements over 8 baselines under both random and class-wise forgetting scenarios.

**Compliance With Llm Reviewing Policy:**

Affirmed.

**Final Justification:**

The author has addressed my concerns, so I will keep my score.

**Key Questions For Authors:**

1. Theorem 3.1 assumes optimal stopping epochs are i.i.d. uniform across samples, an assumption likely violated in practice. How sensitive is the bound to this? Does it hold under more realistic correlation structures?

2. Appendix A shows that linear methods fail disentanglement, but does not explain why the Bézier curve avoids the same problem. What theoretical property of nonlinear interpolation prevents weight entanglement?

3. All experiments are on image classification. Do you expect the MCU to transfer to LLMs, and if so, what loss landscape properties are required?

**Limitations:**

yes

**Strengths And Weaknesses:**

## Strengths


1. Using mode connectivity as an active unlearning mechanism (not just an evaluation tool) is original. The "effective unlearning region" concept, a continuous spectrum of viable models, is a useful contribution that prior work lacks.

2. Masking entire parameter tensors (vs. element-wise) achieves comparable unlearning quality with a 75% speedup. Simple, well-motivated, and practically useful.

3. The evaluation spans 8 baselines, 3 datasets, 3 architectures, and both forgetting scenarios. Figure 15 goes further by applying the same post-hoc selection strategy to baselines, confirming that MCU's gains stem from nonlinear interpolation, not from smarter model selection.


## Weaknesses

1. The i.i.d. uniform assumption on optimal stopping epochs is unrealistic and unvalidated.

2. Appendix A proves linear methods fail disentanglement. But it does not prove or explain why the Bézier curve avoids the same problem. The core motivation is empirical, not theoretical.

3. No experiments or discussion of applicability to language models or other architectures where unlearning is increasingly critical.

---

> ### Author Rebuttal · Authors · 2026-03-27
>
> # **Response to Reviewer whRh**
>
> We thank the reviewer for the constructive feedback and for recognizing the creative contribution of our nonlinear method to machine unlearning. Below, we address the comments in detail.
>
> ### **W1 & Q1: The i.i.d. uniform assumption**
>
> In practice, optimal stopping epochs across samples are likely correlated. We would like to clarify that the i.i.d. uniform assumption is only used to derive a lower bound (of the models needed) for exposition. Our intention is not to provide a tight bound, but to **demonstrate the fundamental insufficiency** of relying on a single model. The key mechanism behind our result is not independence or uniformity, but misalignment of optimal stopping epochs across forgetting samples.
>
> In fact, the theorem can be stated without any i.i.d. assumption. Let $E_m$ denote the event that model $m$ admits a single epoch jointly optimal for all forgetting samples, and define $p_m=\Pr(E_m)$. Then,
>
> $\Pr\!\left(\bigcup_{m=1}^k E_m\right)\ge 1-(1-p_\star)^k,
> \quad p_\star=\inf_m p_m.$
>
> One can see that if we only have one model, $p_\star$ will be close to zero when the number of forgetting data is large, and it is almost impossible to align these data points. To achieve success probability at least $1-\delta$, it suffices that
>
> $k \ge \left\lceil \frac{\ln \delta}{\ln(1-p_\star)} \right\rceil.$
>
> The original result follows as a special case with $p_\star=T^{1-N}$. We will revise the paper to (i) clarify the role of the assumption and (ii) include this dependence-aware formulation.
>
> ### **W2 & Q2: Why the Bézier curve avoids wight entanglement**
>
> Linear methods restrict the solution to a one-dimensional affine subspace, which cannot simultaneously satisfy heterogeneous optimality conditions across samples. This limitation is independent of specific loss functions and arises from insufficient degrees of freedom. Bézier curves address this limitation by introducing additional control points, which expand the feasible set from a line to a higher-dimensional nonlinear manifold, **enlarging the feasible set of candidate solutions in parameter space**. This allows the optimization to navigate toward regions that better balance forgetting and retaining, **rather than being constrained to a fixed linear subspace**.
>
> While we do not claim a theoretical guarantee of full disentanglement for our non-liner method, the empirical evidence is compelling: (1) Table 3 shows that MCU improves the linear baseline NegTV, demonstrating that the nonlinear pathway at least mitigates the entanglement issue; (2) Figure 9 in the Appendix shows that linear interpolation completely fails to produce any effective intermediate unlearning models, compared to our non-linear method as shown in Figure 3.
>
> ### **W3 & Q3: Applicability to language models or other architectures**
>
> Thank you for the insightful suggestion. We will incorporate a discussion on the applicability of our method to language models in the revised paper.
>
> In our work, we mainly focus on image classification following prior works [1,2,3] which also consider only the image classification setting. While we have not yet evaluated MCU on language models, we believe MCU can be extended to language models, and we provide the following discussion.
>
> **1 Theoretical Feasibility**
>
> MCU's core mechanisms are modality-agnostic. Prior work has demonstrated that mode connectivity exists in LLMs [4,5]. Since MCU leverages mode connectivity to construct pathways in parameter space, this foundation naturally extends to language models.
>
> **2 Practical Challenges**
>
> Firstly, LLMs have billions of parameters, requiring careful memory management for storing theta_o, theta_p, and theta_c. Parameter-efficient techniques (e.g., LoRA-style decomposition) maybe needed to mitigate this.
>
> Moreover, transformer components in LLMs may require tailored parameter masking strategies. We may need mechanistic analysis or causal tracing techniques to identify the importance of different modules (e.g., FFN and attention) or layers for forgetting versus retaining data, enabling more targeted masking beyond our current gradient-based approach.
>
> Overall, the current focus on image classification allows us to thoroughly validate MCU's core principles with manageable computational resources, establishing the foundational methodology for mode connectivity-based unlearning. This work serves as an essential stepping stone toward applying MCU to other domains.
>
> ---
>
> **References**
>
> [1] Zhao et al. What makes unlearning hard and what to do about it
>
> [2] Spartalis et al. Lotus: Large-scale machine unlearning with a taste of uncertainty
>
> [3] Georgiev et al. Machine Unlearning via Simulated Oracle Matching
>
> [4] Cheng et al. Understanding Machine Unlearning Through the Lens of Mode Connectivity
>
> [5] Liu et al. Model sparsity can simplify machine unlearning

---

> > ### Author Rebuttal · Reviewer_whRh · 2026-04-04
> >
> > Thank you for the detailed rebuttal, all of my concerns have been addressed. I will keep my score.

---

> > > ### Author Response · Authors · 2026-04-07
> > >
> > > We are very pleased to know that our clarifications have fully addressed your concerns. We sincerely appreciate the reviewer’s time and constructive feedback on our work.

---

### Decision · Program_Chairs · 2026-04-30

**Decision:**

Accept (regular)

**Comment:**

This paper proposes a new approach for machine unlearning using a nonlinear pathway in the parameter space that provides a spectrum of unlearned models. The nonlinear path addresses the weight entanglement that limits that usefulness of linear paths (task arithmetic). An unlearned model can be chosen from the unlearned model spectrum in a computationally efficient way compared to rerunning the unlearning method again if the unlearning goal changes (e.g., the required privacy level for unlearned examples). The proposed approach works in a plug-and-play manner for existing unlearning methods and improves them.

The reviews show appreciation of the proposed idea and method. The weaknesses and questions in the initial reviews were addressed well in the authors’ rebuttal.

The main practical limitation of this paper is the evaluation only for image classification problems and not for larger models such as LLMs and diffusion models.

Therefore, my recommendation is a Weak Accept.